# Mapping DNA damage-dependent genetic interactions in yeast via party mating and barcode fusion genetics

J Javier Díaz-Mejía[1,2,3,4], Albi Celaj[1,2,3,4] (iD), Joseph C Mellor[1,2,3,4,5,†], Atina Coté[1,2,3], Attila Balint[1,6,‡] (iD), Brandon Ho[1,6], Pritpal Bansal[1,2,3], Fatemeh Shaeri[1,2,3], Marinella Gebbia[1,2], Jochen Weile[1,2,3] (iD), Marta Verby[1,3], Anna Karkhanina[1,2,3], YiFan Zhang[1,2,3], Cassandra Wong[3], Justin Rich[1,2,3], D'Arcy Prendergast[1,2,3], Gaurav Gupta[1,2,3], Sedide Öztürk[5,#], Daniel Durocher[2,3], Grant W Brown[1,6] & Frederick P Roth[1,2,3,4,5,7,8,*] (iD)

## Abstract

Condition-dependent genetic interactions can reveal functional relationships between genes that are not evident under standard culture conditions. State-of-the-art yeast genetic interaction mapping, which relies on robotic manipulation of arrays of double-mutant strains, does not scale readily to multi-condition studies. Here, we describe barcode fusion genetics to map genetic interactions (BFG-GI), by which double-mutant strains generated via *en masse* "party" mating can also be monitored *en masse* for growth to detect genetic interactions. By using site-specific recombination to fuse two DNA barcodes, each representing a specific gene deletion, BFG-GI enables multiplexed quantitative tracking of double mutants via next-generation sequencing. We applied BFG-GI to a matrix of DNA repair genes under nine different conditions, including methyl methanesulfonate (MMS), 4-nitroquinoline 1-oxide (4NQO), bleomycin, zeocin, and three other DNA-damaging environments. BFG-GI recapitulated known genetic interactions and yielded new condition-dependent genetic interactions. We validated and further explored a subnetwork of condition-dependent genetic interactions involving *MAG1, SLX4,* and genes encoding the Shu complex, and inferred that loss of the Shu complex leads to an increase in the activation of the checkpoint protein kinase Rad53.

**Keywords** condition-dependent; DNA barcode; *en masse*; genetic interaction; sequencing

**Subject Categories** Genome-Scale & Integrative Biology; Methods & Resources; Network Biology
**Mol Syst Biol.** (2018) 14: e7985

## Introduction

### The importance of condition-dependent genetic interactions

Genetic interactions, defined by a surprising phenotype that is observed when mutations in two genes are combined (Mani *et al*, 2008), are powerful tools to infer gene and pathway functions (Baryshnikova *et al*, 2010; Ideker & Krogan, 2012). Of the genetic interactions currently known in any species, the vast majority were found using Synthetic Genetic Array (SGA) technology in *Saccharomyces cerevisiae* (Bandyopadhyay *et al*, 2010; Costanzo *et al*, 2010, 2016; van Leeuwen *et al*, 2016) and these studies have yielded a rich landscape of genetic interactions. The sign of genetic interaction (defined to be negative when mutants are synergistically deleterious, and positive when the combination is less severe than would be expected from independent effects) provides clues about whether the genes act in parallel or in a concerted or serial fashion. Measuring similarity between genetic interaction profiles, both at the level of single genes and of clusters of genes, has

1 Donnelly Centre, University of Toronto, Toronto, ON, Canada
2 Department of Molecular Genetics, University of Toronto, Toronto, ON, Canada
3 Lunenfeld-Tanenbaum Research Institute, Mt. Sinai Hospital, Toronto, ON, Canada
4 Department of Computer Science, University of Toronto, Toronto, ON, Canada
5 Department of Biological Chemistry and Molecular Pharmacology, Harvard Medical School, Boston, MA, USA
6 Department of Biochemistry, University of Toronto, Toronto, ON, Canada
7 Center for Cancer Systems Biology (CCSB) and Department of Cancer Biology, Dana-Farber Cancer Institute, Boston, MA, USA
8 Canadian Institute for Advanced Research, Toronto, ON, Canada
*Corresponding author. Tel: +1 416 946 5130; E-mail: fritz.roth@utoronto.ca
†Present address: SeqWell, Inc., Beverly, MA, USA
‡Present address: Department of Cellular and Molecular Medicine, Center for Chromosome Stability, University of Copenhagen, Copenhagen, Denmark
#Present address: Roche Sequencing Solutions, Pleasanton, CA, USA

revealed a hierarchical map of eukaryotic gene function (Costanzo et al, 2010, 2016). However, the vast majority of genetic interaction mapping has been conducted under a single standard culture condition.

The importance and qualitative nature of gene function can change with environmental fluctuation, so that a complete understanding of genetic interactions will require mapping under multiple conditions. For example, pairs of DNA repair genes had 2–4 times more genetic interactions between DNA repair genes under MMS treatment compared with rich media alone (St Onge et al, 2007; Bandyopadhyay et al, 2010; Ideker & Krogan, 2012), so that a plethora of condition-dependent genetic interactions remain to be uncovered via gene × gene × environment studies.

## Current genetic interaction discovery technologies

Essentially every large-scale genetic interaction mapping strategy in S. cerevisiae uses a genetic marker system developed for the SGA technique, which works by mating a single-gene deletion query strain with an array of different single-gene deletion strains from the Yeast Knockout Collection (YKO) (Giaever et al, 2002). The SGA system provides genetic markers by which mated diploids can be subjected to a series of selections to ultimately yield haploid double mutants. In "standard" SGA mapping, the fitness of the resulting double mutants is determined by statistical analysis of the images from each plate, yielding cell growth estimates for each separately arrayed strain (Tong & Boone, 2005). SGA has also been used to study genetic interactions within functionally enriched gene groups (Collins et al, 2006) and has been applied to detect environment-dependent interactions (St Onge et al, 2007; Bandyopadhyay et al, 2010). For example, St Onge et al (2007) used the SGA markers to generate all pairwise double mutants between 26 DNA repair genes in yeast. The authors cultured each double mutant individually in microplates and monitored cell density over time to infer the fitness of double mutants and thereby identify genetic interactions in the presence and absence of MMS.

Others have measured genetic interactions via competition-based fitness measurements in liquid cultures, adding fluorescent markers for tracking cell viability, and using robotic manipulation to inoculate and measure cell growth (DeLuna et al, 2008; Garay et al, 2014). A recent technique called iSeq incorporated barcodes into single-mutant strains, such that pairs of barcodes identifying corresponding pairs of deleted genes could be fused by Cre-mediated recombination (Jaffe et al, 2017). The authors demonstrated the method, showing that a pool corresponding to nine gene pairs could be sequenced to monitor competitive growth of double mutants en masse in different environments (Jaffe et al, 2017). Cre-mediated approaches have been used similarly to map protein–protein interactions (Hastie & Pruitt, 2007; Yachie et al, 2016; Schlecht et al, 2017).

For each of the above genetic interaction methods, double mutants were generated by individual mating of two specific yeast strains, requiring at least one distinct location for each double-mutant strain on an agar or microwell plate and necessitating robotic strain manipulation to achieve large scale. By contrast, other methods to map genetic interactions generated double mutants in a "one-by-many" fashion. For example, diploid-based synthetic lethality analysis on microarrays (dSLAM) (Pan et al,

2004) disrupted a single "query" gene by homologous recombination via transformation of a marker into a pool of diploid heterozygous deletion strains bearing the SGA marker. After selecting for double-mutant haploids from such a one-by-many haploid double-mutant pool, barcodes were PCR-amplified from extracted double-mutant DNA and hybridized to microarrays to infer the relative abundance and fitness of each double mutant. Another method, genetic interaction mapping (GIM) (Decourty et al, 2008), generated a one-by-many pool of barcoded double mutants by en masse mating a single query strain to a pool of haploid gene deletion strains. Like dSLAM, GIM inferred strain abundance and fitness via barcode hybridization to microarrays. Despite the efficiency of generating one-by-many double-mutant pools, a matrix involving thousands of query strains would require thousands of such pools to be generated.

Each of the above methods has advantages and disadvantages. For example, measuring a growth time-course for each double-mutant strain provides high-resolution fitness measurements (St Onge et al, 2007; Garay et al, 2014), but scalability is low. Standard SGA is high-throughput, but requires specialized equipment for robotic manipulation, and these manipulations must be repeated to test genetic interactions in new environments. The iSeq method shares the scaling challenge of SGA in strain construction, in that it requires many pairwise mating operations; however, once a double-mutant pool has been generated, it represents a promising strategy for measurement of competitive pools in different environments. The dSLAM and GIM methods allow generation of one-by-many pools, which reduces the number of mating operations, but both methods require customized microarrays as well as pool generation and microarray hybridization steps for every query mutation in the matrix.

## Barcode fusion genetics to map genetic interactions (BFG-GI)

Here, we describe BFG-GI, which borrows elements from several previous approaches. Like iSeq, BFG-GI requires generation of barcoded single-mutant strains, with only minimal use of robotics. To generate double-mutant pools, BFG-GI uses the SGA marker system and, like the GIM strategy, BFG-GI employs en masse mating. Unlike GIM and all other previous genetic interaction mapping strategies, BFG-GI employs many-by-many "party mating" to generate all double mutants for a matrix of genes in a single mating step. All successive steps—including barcode fusion, sporulation, selection of haploid double mutants, and measurement of relative strain abundance—are also conducted en masse. We show that double mutants can be generated and monitored in competitive pools using BFG-GI. Like iSeq, BFG-GI infers double-mutant fitness in competitively grown strain pools using next-generation sequencing of fused barcodes, and BFG-GI double-mutant pools can be aliquoted and stored. Aliquots can be thawed later and challenged under specific environments (e.g., drugs) to detect condition-dependent genetic interactions without having to regenerate the double-mutant strains.

We assessed BFG-GI by mapping genetic interactions of DNA repair-related genes under multiple DNA-damaging conditions, revealing many condition-dependent interactions and a discovery that perturbation of the Shu complex leads to increased activation of the Rad53 checkpoint protein kinase.

# Results

## BFG-GI experimental design overview

The first step in the BFG-GI process is generating uniquely barcoded donor and recipient strains with complementary mating types. Each donor and recipient strain contains a unique barcode locus. In the donor strain, this barcode is flanked by two distinct site-specific recombination sites (*loxP/2272* sites), while in the recipient strain, both recombination sites lie on the same side of the unique recipient barcode. After the mating step, these sites mediate barcode fusion via the Cre/Lox system, yielding chimeric barcode sites that uniquely identify specific deletion combinations. We created donors by crossing individual gene deletion strains from the YKO collection with proDonor strains that contained newly constructed pDonor plasmids (Figs 1A and EV1, and Materials and Methods). We generated recipient strains by crossing individual gene deletion strains from the SGA query collection with proRecipient strains (Figs 1B and EV2, and Materials and Methods). Haploid selection of double mutants followed mating of donor and recipient strains, sporulation, and *in vivo* fusion of barcodes using Cre/Lox recombination (Fig 1C).

We confirmed that barcode fusion was successful using two control strains carrying markers at likely-neutral loci. Specifically, we crossed a *MAT*alpha Donor *hoΔ::kanMX* to a *MAT***a** Recipient *ylr179cΔ::natMX* and induced Cre/Lox recombination to fuse their barcodes. After sporulation and selection of the *MAT*alpha haploid double-mutant progeny (Materials and Methods), we extracted genomic DNA, amplified barcode fusions by PCR, and confirmed their integrity by Sanger sequencing (Fig 1C).

To scale up the BFG-GI process, we optimized mating and sporulation steps to generate double mutants with unique barcodes that had been fused *en masse* (Materials and Methods). We selected hundreds of double mutants using a series of marker selection steps in a many-by-many fashion. Intermediate selection steps allowed us to fuse barcodes representing each donor and recipient parental pair within each double-mutant cell (Fig 1D and Materials and Methods).

Once we generated the pool of fused-barcode double mutants, aliquots were stored at −80°C for future experiments. Amplification and next-generation sequencing of fused barcodes in the pool allowed us to infer the relative abundance of each double mutant in each condition of interest (Fig 1D and Materials and Methods). In addition to haploid double-mutant pools, we sequenced fused barcodes from the heterozygous diploid double-mutant pools and used those as reference ("time zero") controls for fitness and genetic interaction calculations (Materials and Methods).

## BFG-GI measures strain abundances within a heterogeneous population

We first evaluated the ability of BFG-GI to accurately detect the abundance of pooled double-mutant strains. To generate reference data for this evaluation, we used the array-based SGA strategy to generate 2,800 double mutants by individual mating of barcoded BFG-GI strains, subsequently inducing barcode fusion via the Cre/Lox system. The purpose of this experiment was to assess the extent to which quantifying growth via fused-barcode sequencing of pooled strains could recapitulate the measurements of growth in individual cell patches (as in conventional SGA). We recorded patch sizes, scraped plates to pool all double-mutant cells, extracted genomic DNA, and sequenced the fused barcodes (Materials and Methods). The resulting numbers of sequencing reads for each strain were strongly correlated with the corresponding colony sizes ($r = 0.92$; Fig 2A). Importantly, colonies that were very small or absent often corresponded to double mutants with very few or no sequencing reads. These results show that BFG-GI detects the abundance of specific double mutants in pools of cells, with results comparable to an array-based method.

## Generating a DNA repair-focused double-mutant strain pool

To test whether BFG-GI can accurately map genetic interactions, we generated a double-mutant pool focused on DNA repair genes and compared BFG-GI results to those of other validated genetic interaction assays. We began by generating donor and recipient strains by crossing 35 YKO (*yfg1Δ::kanMX, MAT***a**) single-gene deletion strains to 65 BFG-GI proDonor strains, and 38 SGA query (*yfg2Δ::natMX, MAT*alpha) single-gene deletion strains to 71 BFG-GI proRecipient strains. The set of deleted genes to which these strains correspond include 26 DNA repair genes from a previous condition-dependent genetic interaction study (St Onge *et al*, 2007), as well as 14 likely neutral loci (i.e., the already-disrupted *HO* locus, pseudogenes, and other loci for which single- and double-mutant phenotypes have not been previously observed). Inclusion of neutral loci allowed us to infer single-mutant fitness from pools of double mutants (Materials and Methods).

To generate haploid double mutants, donor and recipient cells were scraped from plates and all subsequent steps in the BFG-GI pipeline were conducted *en masse*. First, the pools were combined for party mating. Seven selection steps followed mating, including four that correspond to those in the standard SGA procedure: heterozygous diploid selection, sporulation, *MAT***a** progeny selection, and haploid double-mutant selection. Additionally, before sporulation, we completed three selection steps to fuse barcodes and subsequently remove Cre to limit additional recombination events (Figs 1C and EV3). This generated a pool of 4,288 haploid double mutants, which was aliquoted and stored as frozen glycerol stock. Thawed samples were used to inoculate solid media appropriate for selecting haploid double-mutant cells. The media was used alone, supplemented with dimethyl sulfoxide (DMSO) as a solvent control, or supplemented with one of seven drugs targeting DNA repair pathways (Table EV1). We extracted genomic DNA and amplified and sequenced fused barcodes to infer the relative abundance of each double mutant in each condition.

To evaluate assay reproducibility, we ran all BFG-GI procedures in duplicate, starting from the mating step (technical replicates) and also barcoded multiple strains representing the same gene (biological replicates). Biological replicate strains had either the same or different parental strain origin (the parental strain for a given gene deletion might be from either the YKO or SGA query strain collection). Relative strain abundance was highly correlated between technical replicates ($r > 0.95$). Next, we used a multiplicative model to infer a genetic interaction score (*GIS*) from relative strain abundances, analogous to other methods based on strain growth (Materials and Methods). The relative strain abundance, *GIS* correlation between technical replicates was also high ($r = 0.96$).

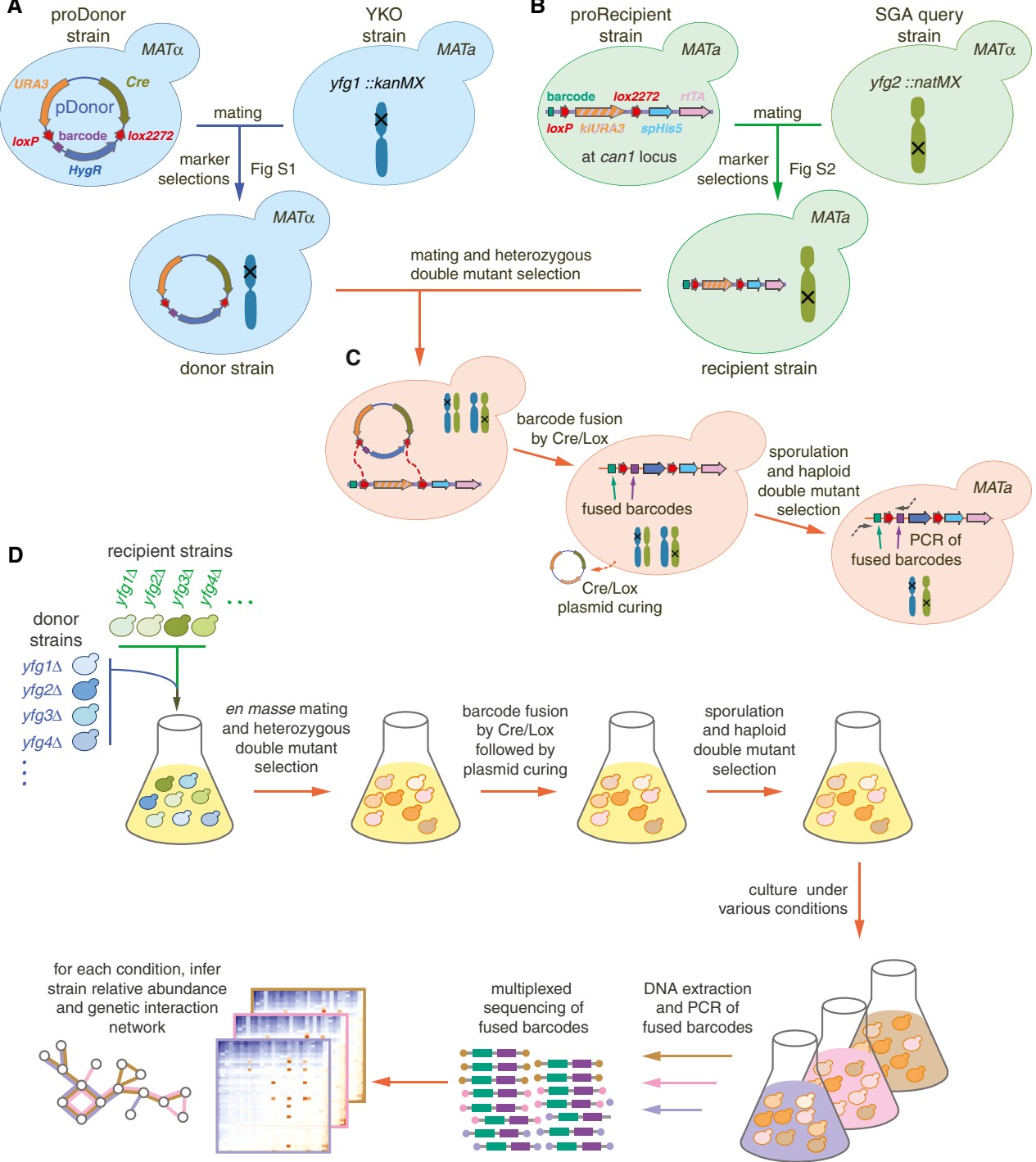

**Figure 1. BFG-GI pipeline summary.**

A   Construction of donors with unique barcodes representing each gene deletion in parental strains from the YKO collection.

B   Construction of recipients also with unique barcodes representing genes of interest in parental strains from the SGA query collection. Pairs of recombination sites (loxP and lox2272) were located at the barcode loci of donor and recipient strains to enable *in vivo* intracellular fusion of barcode pairs at the recipient barcode locus.

C   Donors and recipients were mated with each other to generate heterozygous diploid double mutants, and barcodes were fused *in vivo* by the Cre/Lox system. The relic plasmid remaining in donors after Cre/Lox recombination was counter-selected after barcode fusion. Sporulation was induced to select for the *MAT***a** progeny and haploid double mutants.

D   BFG-GI was conducted *en masse* to generate "many-by-many" pools for a set of 26 DNA repair and 14 neutral genes. The resulting pool of haploid double mutants was stored as aliquots of glycerol stock. Thawed aliquots were used to inoculate media containing different chemical agents ("drugs"). Genomic DNA was extracted and fused barcodes were amplified and sequenced to monitor double-mutant abundance and to infer genetic interactions. Details of donor and recipient strain construction are shown in Figs EV1 and EV2, respectively. Media details are shown in Fig EV3.

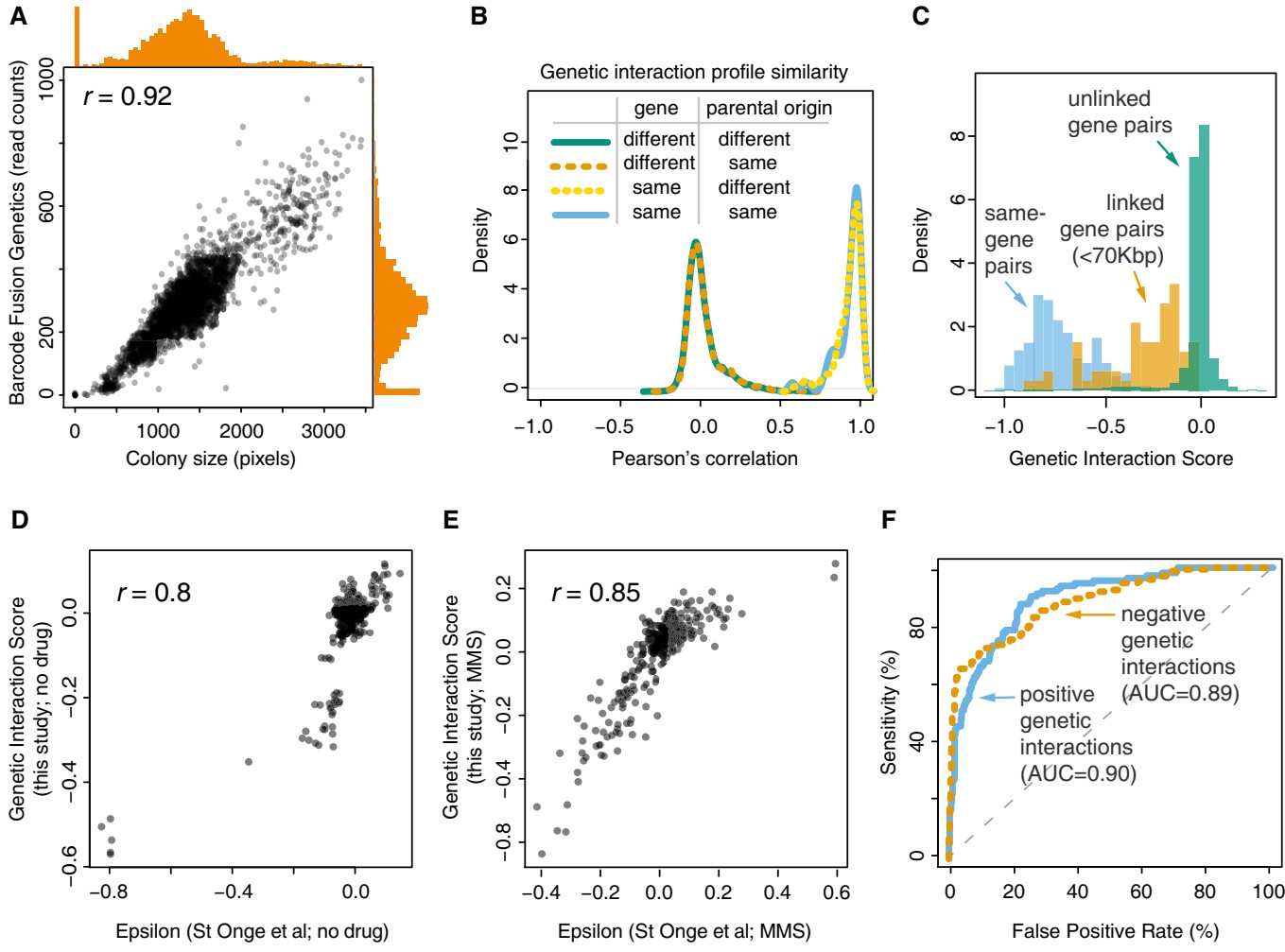

**Figure 2. BFG-GI quality control and benchmarking.**

A   Correlation between two measures of cell abundance (colony size and next-generation-sequencing-based quantification of fused barcodes) for BFG-GI double-mutant strains. Histograms show distribution of abundance in the two measurements. Peaks in the histograms representing data points in the bottom-left corner of the scatter plot indicate that absent and very small colonies produced few or no sequencing reads.

B   Density plots for BFG-GI genetic interaction score (*GIS*) correlation between replicates of the same gene, with same or different parental origin, or pairs of different genes. Only replicates with a *GIS* correlation > 0.5 were retained for further analyses.

C   Histograms comparing the *GIS* distribution for "same-gene pairs" (which are expected to behave like synthetic lethals given the SGA double-mutant selection process) with that for linked- and unlinked-gene pairs.

D   Comparison of BFG-GI-inferred genetic interactions in haploid double-mutant media without MMS with genetic interactions identified using similar media (St Onge *et al*, 2007).

E   Comparison of BFG-GI-inferred genetic interactions in haploid double-mutant media containing MMS with genetic interactions previously identified in similar media (St Onge *et al*, 2007).

F   Benchmarking of BFG-GI genetic interactions against the St. Onge *et al* (2007) dataset. Note that "false positives" may be real interactions that were not found in the benchmark study.

Correlation of *GIS* profiles between biological replicates representing the same gene was generally high, with 85% of replicates showing *GIS r* > 0.5. We computationally excluded from analysis 21 biological replicates (six donors and 15 recipients) showing *GIS r* < 0.5. For the remaining strains, biological replicate profiles clearly showed higher correlation than did the profiles of replicates carrying deletions in different genes (Fig 2B). To understand factors contributing to poorly correlated replicate pairs, we sequenced the genomes of 20 strain pairs. Ten of those pairs corresponded to strains with *GIS r* < 0.5 and other 10 with *GIS r* > 0.5. We found that all 10

strain pairs with *GIS r* < 0.5 had chromosome V duplicated in one of the two strains, in agreement with the report of iSeq strains showing low strain profile reproducibility, owing to this same chromosome V duplication (Jaffe *et al*, 2017). Chromosome V contains the *CAN1* locus, the locus at which both BFG-GI recipients and iSeq strain constructs are inserted. By contrast, only three out of 10 strain pairs with *r* > 0.5 showed aneuploidies in just one strain in the pair (for these strains, the aneuploidies were also in chromosome V). All BFG-GI strains showing aneuploidies were recipients. This suggests that future versions of BFG-GI recipients for which selection markers are

carried by plasmids may increase reproducibility, as we found for our Donor strains. Furthermore, we removed strains with poor representation in the heterozygous diploid pool, because *GIS* profiles from these strains yielded neutral scores even for controls ("same-gene" pairs described below) that should behave like strong negative interactions, presumably due to poor statistical power to detect fitness effects (Fig EV4B). This included all replicates representing *swc5Δ*, which showed very low relative abundance in the sequencing results. Our final dataset consisted of 3,232 double mutants, with 59 Donors and 56 Recipients, representing 39 genes (25 DNA repair genes and 14 neutral genes; Fig EV4A and Table EV2). Finally, *GIS* measurements for technical and biological replicates (Table EV3) were combined into a single score for each gene pair (Table EV4; Materials and Methods).

We next assessed the ability of BFG-GI to infer the fitness for three classes of double-mutant strains. First, we measured the abundance of strains carrying two differently barcoded mutations corresponding to the same gene. Compound heterozygous diploids bearing a mutation at both loci for a given gene (e.g., *mms4Δ:: kanMX/mms4Δ::natMX*) can survive in media supplemented with selective antibiotics; however, haploid cells derived from this parental diploid should not survive because they should only carry one locus for each gene and therefore only one of the two antibiotic resistance markers required to survive the selection. Thus, haploid strains for "same-gene pairs" are expected to exhibit reduced fitness, behaving like synthetic lethal combinations, and be depleted from the pools. The calculated *GIS* agreed with this expectation (Fig 2C). Second, we assessed the abundance of double mutants representing pairs of linked genes (< 75 kbp apart; Fig EV4C). Independent segregation is reduced between linked genes, and as expected, our *GIS* indicated these double mutants were also depleted from the pools (Fig 2C). Third, we analyzed double mutants representing unlinked genes and we found that their *GIS* distribution is clearly distinguishable from same-gene and linked-gene pairs (Fig 2C).

Finally, we sought to compare BFG-GI results against another dataset of genetic interactions (St Onge *et al*, 2007), both to obtain an overall evaluation of our method and as a way to calibrate our *GIS* thresholds for calling genetic interactions. We first compared BFG-GI *GIS*s with the epsilon scores reported by St Onge *et al* (2007) under both no-drug and MMS conditions, for pairs of DNA repair genes that had been tested in both studies. We found that *GIS* and epsilon scores correlated well with each other in both no-drug ($r = 0.8$) and MMS ($r = 0.85$) conditions (Fig 2D and E). Taking both conditions together, and using *GIS* thresholds with an estimated 5% false-positive rate, BFG-GI captured 56% of the positive genetic interactions reported by St. Onge *et al* and 66% of the negative genetic interactions (Fig 2F), while reporting an

additional 23 positive and 20 negative interactions not reported by St Onge *et al* (2007).

Taken together, these results provide evidence that BFG-GI offers a powerful means of generating double mutants by *en masse* mating and monitoring strain abundance in a multiplexed fashion to infer condition-dependent genetic interactions.

## BFG-GI reveals condition-dependent genetic interactions

Having determined that BFG-GI can accurately detect genetic interactions, we analyzed the same double-mutant pool under seven additional culture conditions to more broadly explore condition-dependent genetic interactions (see Fig 3C legend for condition names and Table EV1 for details). To call positive and negative interactions, we first standardized *GIS* by the estimated error ($Z_{GIS}$; Materials and Methods), and used the distribution of $Z_{GIS}$ amongst unlinked barcode pairs containing a neutral gene ("neutral pairs"; Fig EV4A) to estimate the false discovery rate (FDR) at each given $Z_{GIS}$ cutoff (Fig EV4D–E). To call interactions, we used both a $Z_{GIS}$ cutoff corresponding to FDR = 0.01 in each condition and an additional effect-size cutoff ($|GIS| > 0.075$) to filter out interactions of high confidence but low magnitude. At these cutoffs, 91% of the called negative interactions and 77% of the called positive interactions were also observed in a previous study (St Onge *et al*, 2007), while 64% of the previously reported negative and 44% of the previously reported positive interactions were reproduced by BFG-GI (Fig EV4F; Table EV4).

Analyzing BFG-GI results further, we found that all DNA repair genes showed at least one genetic interaction and that some genes showed markedly more interactions than others. For example, we found that the DNA helicase gene *SGS1* yielded negative interactions with *MMS4, MUS81*, or *SLX4* (all of which participate in template switching during break-induced replication) in all nine conditions (Fig 3A, Table EV4). Another DNA helicase gene, *SRS2*, interacted negatively with both *SGS1* and the DNA translocase gene *RAD54* in all nine conditions. A third DNA helicase/ubiquitin ligase gene, *RAD5,* showed positive genetic interactions with *SGS1* in six conditions. *SGS1* and *SRS2* are involved in error-free DNA damage tolerance, while *RAD5* is involved in recombinational repair of double-strand breaks. These findings coincide with previous reports showing *SGS1* and *SRS2* centrality in DNA repair pathways in both unperturbed and MMS-induced stress conditions (St Onge *et al*, 2007).

We next examined condition-dependent changes in genetic interactions. First, genetic interaction differences between conditions were calculated (Δ*GIS*). Then, using a similar approach to that which was used to call genetic interactions within each condition,

**Figure 3. Condition-dependent genetic interactions mapped by BFG-GI.**

A   Networks showing the number of conditions with a genetic interaction for each gene pair (using FDR < 0.01 and |*GIS*| > 0.075 as cutoffs). Numbers besides gene names are guides for the reader to locate nodes in networks of panels (B) and (C). Data for individual interactions are available in Tables EV3 and EV4.

B   Networks in the diagonal (subpanels *ii* and *iii*) show genetic interactions for DMSO or MMS after applying the same criteria as in (A). The network in subpanel *i* shows significant genetic interaction changes (FDR < 0.01, |Δ*GIS*| > 0.1) when comparing the DMSO and MMS treatments. Interaction types are positive (+), negative (−), or neutral (n). The barplot in subpanel *iv* summarizes the number of changes between interaction type in subpanel *i*.

C   The networks are the same as described in (B) with additional drug conditions: cisplatin (CSPL), doxorubicin (DXRB), hydroxyurea (HYDX), zeocin (ZEOC), bleomycin (BLMC), and 4NQO. The no-drug condition was omitted from this figure as it showed no significant condition-dependent genetic interactions with DMSO. *GIS* profiles were hierarchically clustered using maximum distance and complete linkage, with the resulting dendrogram shown on the left. Data for individual differential interactions are available in Tables EV5 and EV6. This figure was generated with Cytoscape (Shannon *et al*, 2009) and R scripts (R Core Team, 2017).

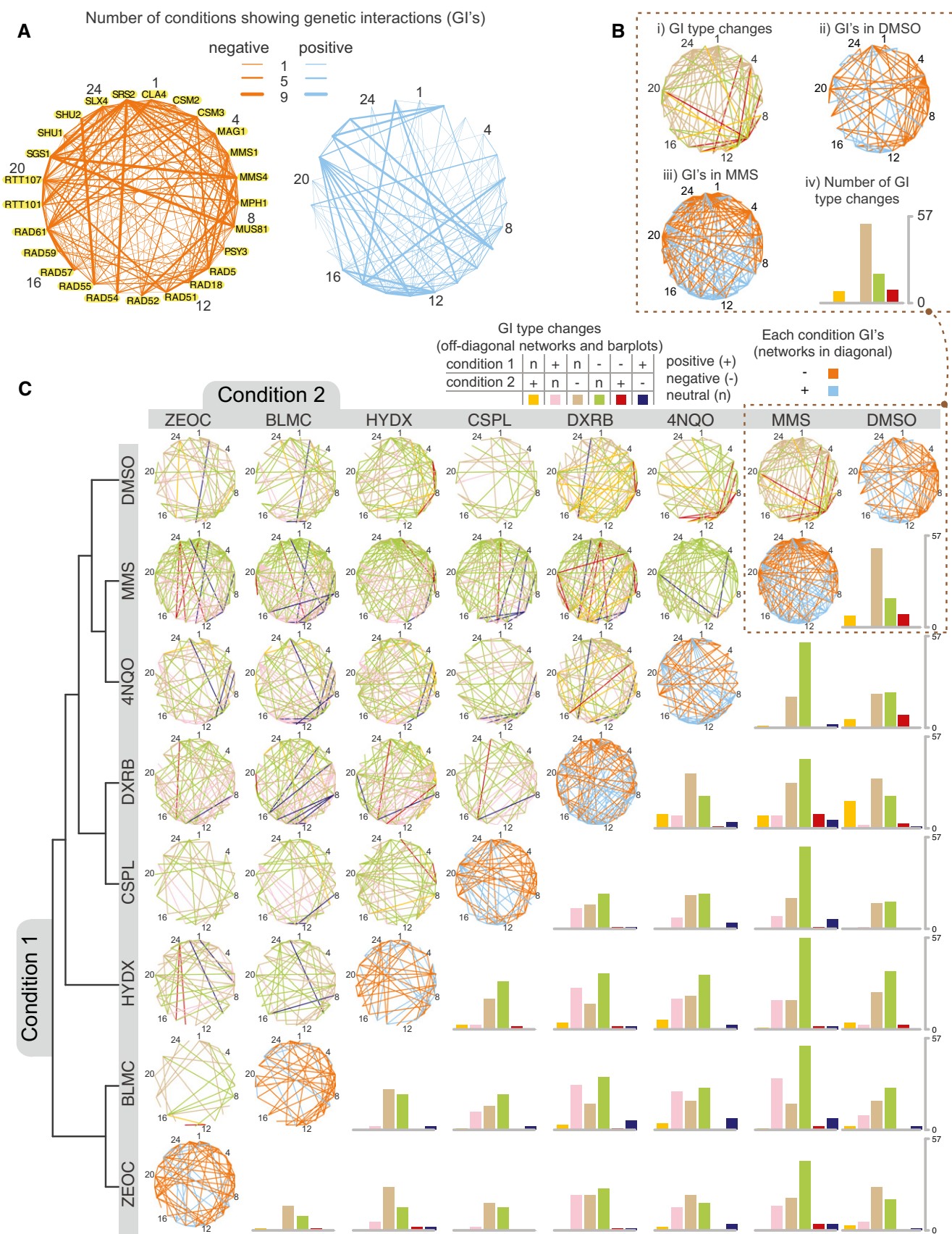

**Figure 3.**

$\Delta GIS$ was standardized by the estimated error ($\Delta Z_{GIS}$), and the distribution of $\Delta Z_{GIS}$ amongst neutral pairs was used to calculate an FDR for each differential interaction (Fig EV5A; Materials and Methods). At a $\Delta Z_{GIS}$ cutoff corresponding to FDR = 0.01 and an effect-size cutoff of $|\Delta GIS| > 0.1$, we identified 2,932 differential interactions amongst DNA damage genes and further considered only the subset of 2,335 differential interactions that changed between interaction type (i.e., between the three classes of positive, negative, and neutral) for further analysis. For any given pair of conditions, an average of 9% of all gene pairs exhibited differential interaction. For example, we found $mus81\Delta/rad5\Delta$ displayed a negative genetic interaction in DMSO, a positive genetic interaction in MMS, and a significant difference between the two conditions. This change is shown as a red edge in Fig 3B, panel $i$, and agrees with a previous report (St Onge $et al$, 2007). By contrast, most changes in genetic interaction between DMSO and MMS were from neutrality in one condition to either a positive or negative genetic interaction in the other (Fig 3B, panels $i$ and $iv$). Generalizing this observation to all pairwise condition comparisons, a large majority of significant differential genetic interactions were neutral in one condition and either positive or negative in the other (94%), and thus, only 6% of significant genetic interactions changed sign between conditions (Fig 3C and Table EV5).

Genes differed both in the total number of differential genetic interactions in which they participated (Fig EV5B) and in the number of their differential genetic interactions that involved a change in sign (Fig EV5C). Genetic interactions involving $RAD5$ were especially dynamic—$RAD5$ participated in 233 significant differential genetic interactions (out of 1,224 comparisons; Fig EV5B), and 55 of these involved sign reversals (Fig EV5C). Out of 55 sign-reversed differential genetic interactions involving $RAD5$, 48 involved $MMS4$, $MUS81$, $RAD51$, $RAD54$, or $RAD55$ (Fig EV5D). $MUS81$ and $MMS4$ encode a heterodimer which cleaves nicked intermediates in recombinational DNA repair (Schwartz $et al$, 2012), while $RAD51$ binds ssDNA to facilitate homologous recombination and requires $RAD54$ and $RAD55$ for its activity (Sugawara $et al$, 2003). Genetic interactions with $RAD5$ were often positive for all five of these genes in 4NQO and MMS, and negative with all five in other tested conditions (Fig EV5D). These findings are consistent with previously reported negative interactions of $RAD5$ with these genes in MMS and positive interactions when no drug stress is added (St Onge $et al$, 2007; Table EV4). The dynamic interactions of $RAD5$ with these two gene groups may reflect the previously reported multifunctional nature of $RAD5$ and its ability to coordinate repair events and replication fork progression differently in response to different types of lesions (Choi $et al$, 2015).

We assessed similarity between growth conditions as measured by similarity between patterns of $GIS$ profiles. As expected, the two conditions most similar to each other were no-drug and DMSO, which also yielded no significant between-condition differential interactions (Table EV5). A hierarchical clustering of conditions by their $GIS$ profiles (Fig 3C) showed that pairs of drugs with similar mechanisms of action clustered together. For example, bleomycin and zeocin, which are members of the same family of glycopeptides that intercalate into DNA to induce double-strand breaks (Claussen & Long, 1999), were grouped as nearest neighbors and also had the least number of differential interactions between any two drug pairs (26, compared to an average of 67 across all condition pairs).

Interestingly, MMS and 4NQO were also grouped as nearest neighbors. Although there were a large number of differential interactions between them (75), the vast majority (73) showed neutrality in one condition and negative genetic interaction in the other. MMS and 4NQO are members of different drug classes, but both are DNA alkylating agents (Xiao & Chow, 1998; Svensson $et al$, 2012). Both MMS and 4NQO cause checkpoint-modulated fork stalling (Minca & Kowalski, 2011; Iyer & Rhind, 2017) that appears to facilitate replication of damaged templates allowing forks to quickly pass lesions (Iyer & Rhind, 2017). Furthermore, strains carrying deletion of genes involved in postreplication repair (PRR) processes, such as MMS2, RAD5, and UBC13, are significantly hypersensitive to both MMS and 4NQO (Lee $et al$, 2014), suggesting that PRR acts on both MMS and 4NQO lesions. DNA lesions caused by these drugs are typically corrected by either base-excision repair (MMS) or nucleotide-excision repair (4NQO), and these pathways are synergistic with each other in genetic backgrounds like $mag1\Delta$ (Xiao & Chow, 1998). We believe that these mechanistic similarities between MMS and 4NQO contributed to the similarity between their GIS profiles in comparison with those from other drugs we tested.

The most divergent condition pairs (those yielding the highest number of differential interactions) were MMS versus doxorubicin (104 changes) and MMS versus bleomycin (110 changes). These results are consistent with the fact that MMS, doxorubicin, and bleomycin have different mechanisms of action and cause DNA lesions that are repaired by different pathways.

### A condition-dependent subnetwork of $MAG1$, $SLX4$, and Shu complex genes

The Shu complex (a heterotetrameric protein complex consisting of Csm2, Psy3, Shu1, and Shu2) promotes Rad51 filament formation and homologous recombination during error-free lesion bypass, double-strand break repair, and meiosis (Mankouri $et al$, 2007; Ball $et al$, 2009; Bernstein $et al$, 2011; Godin $et al$, 2013; Sasanuma $et al$, 2013) (Fig 4A). Our BFG-GI results indicated that genes encoding all four members of the Shu complex showed negative genetic interactions with both $MAG1$ and $SLX4$ during exposure to MMS. Additionally, the Shu complex genes interacted negatively with $SLX4$ during treatment with 4NQO, bleomycin, and zeocin (Fig 4B). Mag1 is a 3-methyladenine DNA glycosylase that removes alkylated bases from DNA to initiate base-excision repair (BER), thereby protecting cells against alkylating agents like MMS (Berdal $et al$, 1990; Chen $et al$, 1990). Slx4 promotes the activity of three structure-specific endonucleases (Fricke & Brill, 2003; Flott $et al$, 2007; Toh $et al$, 2010; Gritenaite $et al$, 2014) and, upon exposure to MMS, plays a key role in down-regulating phosphorylation of the checkpoint kinase Rad53 (Ohouo $et al$, 2013; Jablonowski $et al$, 2015). We generated double mutants for each Shu complex member in combination with either $MAG1$ or $SLX4$ and tested fitness on media containing DMSO or various genotoxins using spot dilution assays (Fig 4C). Our results validated the $MAG1$–Shu complex gene interactions in MMS that we detected with BFG-GI, and are consistent with a previous study (Godin $et al$, 2016). The negative interactions between $MAG1$ and Shu complex members are explained (Godin $et al$, 2016) by the fact that these double mutants have simultaneously lost Mag1-mediated BER (which directly removes alkylated bases) and have a diminished capacity for error-free lesion bypass,

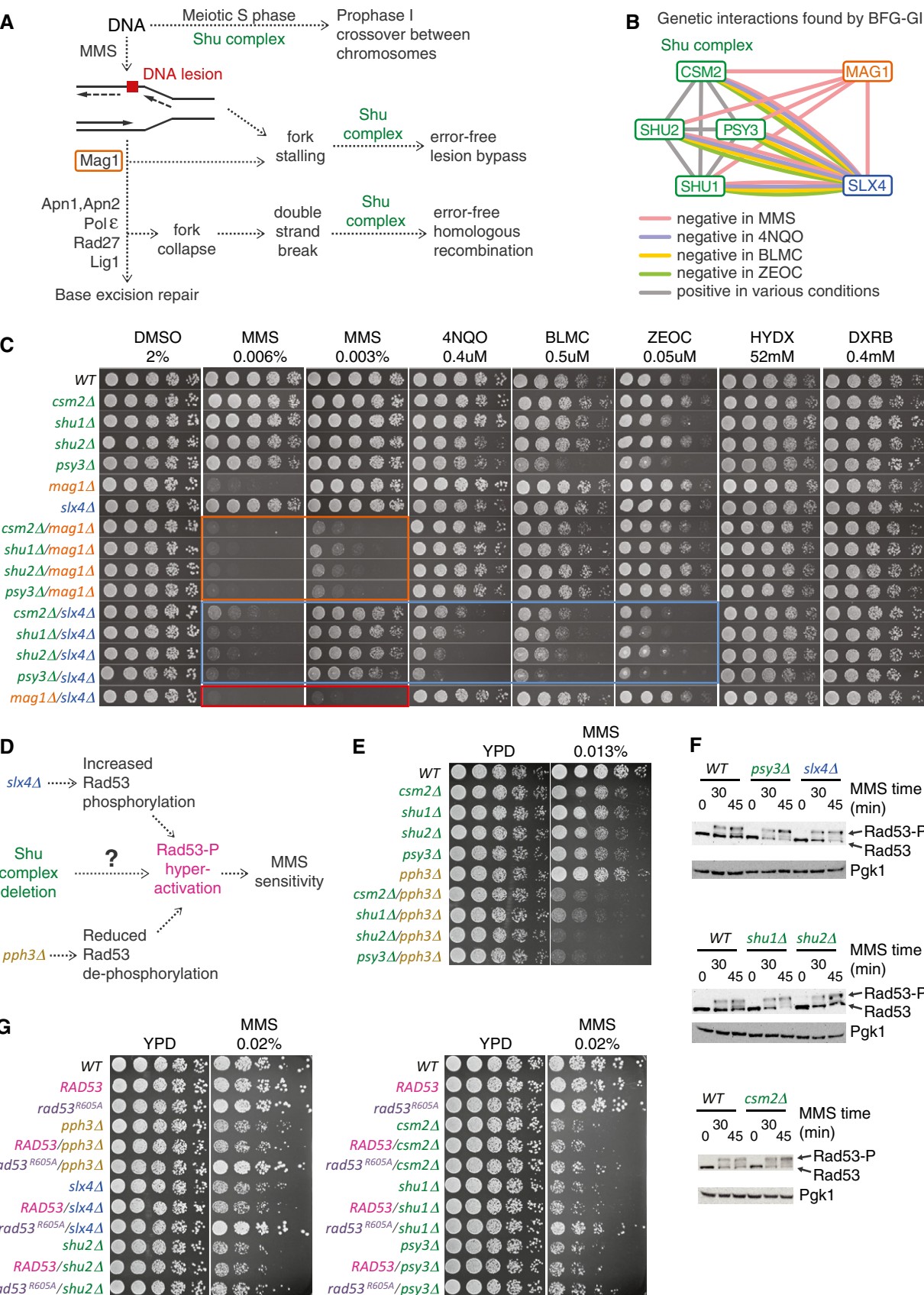

**Figure 4.**

◄

**Figure 4.  Shu complex condition-dependent genetic interactions with *MAG1*, *SLX4*, *PPH3*, and *RAD53*.**

A   Pleiotropic participation of the Shu complex in DNA replication and repair pathways.
B   Network showing condition-dependent genetic interactions inferred from BFG-GI for the indicated conditions.
C   Confirmation of interactions between the Shu complex, *MAG1*, and *SLX4* using spot dilution assays including single and double mutants exposed to the indicated drugs for 48 h. Orange, blue, and red boxes indicate genetic interactions of Shu complex members with *MAG1* and *SLX4*, and of *MAG1* with *SLX4*, respectively.
D   Schematic of potential functional connections between the Shu complex and *SLX4*. As with deletion of *SLX4* or *PPH3*, deletion of Shu complex members may lead to hyperphosphorylation and hyperactivation of Rad53, resulting in increased sensitivity to MMS.
E   Spot dilution assays showing genetic interactions of Shu complex genes/*pph3Δ* double mutants and corresponding single mutants exposed to MMS at the indicated concentration for 48 h.
F   Western blot assays showing hyperphosphorylation of Rad53 in *csm2Δ*, *psy3Δ*, *shu1Δ*, and *slx4Δ* strains following treatment with 0.03% MMS. Note increased intensity of Rad53-P bands compared with the Rad53 bands.
G   Spot dilution assays of Shu complex mutants expressing a hypomorphic *rad53-R605A* allele (*rad53-R605A-6xHis-3xFLAG-kanMX6*) compared with a wild-type *RAD53* allele (*RAD53-6xHis-3xFLAG-kanMX6*). Cells were exposed to MMS at the indicated concentration for 60 h.

Source data are available online for this figure.

a major pathway used during MMS-induced blocks in DNA replication (Huang *et al*, 2013) (Fig 4A). Our spot dilution assays also confirmed that *MAG1* interacts negatively with *SLX4* during MMS treatment (Fig 4C). This result is also consistent with a previous study showing that BER is unlikely to be the major function of *SLX4* (Flott *et al*, 2007). Of particular interest, we validated the BFG-GI interactions between Shu complex members and *SLX4* during treatment with MMS, 4NQO, bleomycin, and zeocin (Fig 4C).

As the nature of the *SLX4* interactions with genes encoding Shu complex proteins is unknown, we studied them in more detail. That there are negative genetic interactions between *SLX4* and Shu complex members in MMS was unexpected, given that the Shu complex promotes error-free lesion bypass (Mankouri *et al*, 2007; Ball *et al*, 2009; Xu *et al*, 2013; Godin *et al*, 2016) and *SLX4* is epistatic to genes that regulate error-free lesion bypass during MMS treatment (Flott *et al*, 2007). A major role for Slx4 under MMS conditions is down-regulating phosphorylation and activation of Rad53, which occurs by Slx4 competing with Rad9 for binding to Dpb11 and consequently limiting the formation of Rad9–Dpb11 complexes that activate Rad53 (Pfander & Diffley, 2011; Ohouo *et al*, 2013; Cussiol *et al*, 2015; Jablonowski *et al*, 2015). Levels of phosphorylated Rad53 are also reduced by the presence of *PPH3*, which encodes the catalytic subunit of the protein phosphatase PP4 complex that binds and dephosphorylates Rad53 during MMS treatment (O'Neill *et al*, 2007). Deletions of either *SLX4* or *PPH3* or both genes result in hyperactivation of Rad53 and hypersensitivity to MMS (Jablonowski *et al*, 2015). This phenotype is suppressed by expression of a hypomorphic *rad53-R605A* allele (Ohouo *et al*, 2013; Cussiol *et al*, 2015; Jablonowski *et al*, 2015). To determine whether the genetic interactions between *SLX4* and Shu complex members (Fig 4C) reveal an unanticipated role for the Shu complex regulating activation of Rad53 (Fig 4D), we tested the sensitivity of *pph3Δ*/Shu complex double mutants to MMS using spot dilution assays. Combining *pph3Δ* with deletion of any of the Shu complex genes resulted in a dramatic increase in MMS sensitivity relative to the single mutants (Fig 4E), indicating negative genetic interactions similar to those seen between *SLX4* and Shu complex members (Fig 4C), or between *SLX4* and *PPH3* (Jablonowski *et al*, 2015).

To assess MMS-induced Rad53 activation in Shu complex mutants more directly, we monitored Rad53 phosphorylation (which is a proxy for Rad53 activation) using Western blot assays. Consistent with the role of *SLX4* in dampening Rad53 activation (Ohouo

*et al*, 2013; Balint *et al*, 2015; Jablonowski *et al*, 2015), *slx4Δ* cells challenged with MMS showed an increase in Rad53-P levels relative to wild type (Fig 4F). Interestingly, three of the Shu complex mutants (*csm2Δ*, *psy3Δ*, and *shu1Δ*) also showed an increase in Rad53-P levels upon treatment with MMS (Fig 4F), indicating that these Shu complex mutants, like *slx4Δ* and *pph3Δ* cells, display hyperactivated Rad53 under exposure to MMS. We asked whether the MMS sensitivity of Shu complex mutants could be suppressed by expression of the *rad53-R605A* allele. Expression of *rad53-R605A*, which is not effectively hyperactivated, suppresses the MMS sensitivity of *slx4Δ* and *pph3Δ* (Ohouo *et al*, 2013; Jablonowski *et al*, 2015). Similarly, the MMS sensitivity of *csm2Δ*, *psy3Δ*, *shu1Δ*, and *shu2Δ* mutants was partially suppressed by *rad53-R605A* (Fig 4G). Together, our data indicate that deletions of genes encoding the Shu complex, as for Slx4 and Pph3, lead to an increase in Rad53 activation, in response to MMS treatment, as revealed by unique condition-dependent genetic interactions detected by BFG-GI.

## Discussion

We developed a new technology, called BFG-GI, in which pools of double-mutant yeast strains corresponding to a matrix of target genes are generated *en masse* through many-by-many "party" mating. These pools are induced to form double-mutant-identifying chimeric barcodes by intracellular site-specific recombination and assayed for growth via next-generation sequencing. Aliquots of these pools can be stored and later cultured with different drugs to identify condition-dependent genetic interactions. To our knowledge, BFG-GI is the first method to generate haploid double-mutant strains *en masse* for a many-by-many matrix of genes without the requirement for multiple mating steps, thus enabling large-scale conditional genetic interaction mapping without extensive use of robotics.

BFG-GI showed good agreement with a previous genetic interaction mapping method (St Onge *et al*, 2007). Quantitatively, our *GISs* show a correlation of $r = 0.8–0.85$ with the epsilon scores obtained in St Onge *et al* (2007). Considering only significant interactions, 91% of the negative and 77% of the positive interactions found by BFG-GI were also observed by St Onge *et al* (2007), and 44–64% of St Onge *et al* (2007) interactions were reproduced by BFG-GI. The contrast between the 0.01 FDR estimate and the validation rate by an orthogonal method suggests that the latter is a too-conservative measure of precision and that many of the novel interactions are

bona fide interactions despite not having been seen by St Onge *et al* (2007).

We detected and validated unanticipated interactions between *SLX4* and Shu complex genes, which mirrored the genetic interactions observed between *PPH3* and the Shu complex. We further found that presence of a functional Shu complex corresponded to reduced activation of Rad53 during MMS treatment.

By calculating similarity between the genetic interaction profiles of different drugs, we found that those with similar mechanisms of action, like zeocin and bleomycin, are considerably more alike than comparisons between compounds with different mechanisms of action, e.g., the comparison between MMS and either zeocin or bleomycin. This suggests the potential of BFG-GI to shed light on drug mechanisms through measurement of gene–gene–environment interactions.

One advantage of BFG-GI is its cost-effectiveness. BFG-GI uses fewer reagents and less robotic assistance than other technologies to map genetic interactions. Like other pool-based technologies, BFG-GI requires less media, plates, and drugs than array-based technologies, resulting in a substantial cost advantage. For example, the amount of media used in 1,536 spot arrays on OmniTrays is reduced 50-fold by studying the same number of gene pairs in pooled cultures with $4 \times 10^6$ cells/cm$^2$ in 143-cm$^2$ Petri dishes, which is the optimal cell density we calculated for pooled double-mutant selections (Materials and Methods). BFG-GI is also more cost-effective than other barcode-sequencing technologies because in BFG-GI, strains are pooled before the mating step, so that generating double mutants does not require robotic manipulation of strain arrays.

The reproducibility of BFG-GI indicates that it is a robust technology. Technical replicates in BFG-GI are highly reproducible, and 85% of the biological replicates correlated well with each other (*GIS* $r > 0.5$). The remaining 15% of biological replicates showing low correlations could be identified and removed computationally. We concur with the iSeq study (Jaffe *et al*, 2017) that aneuploidies in chromosome V are the main factor contributing to the replicates with low reproducibility. Chromosome V carries both *CAN1* and *URA3* loci, which were replaced by selection markers in the iSeq protocol (Jaffe *et al*, 2017), while *CAN1* was replaced by the recipient constructs in BFG-GI. Thus, *de novo* structural variation around these loci during strain construction could explain the low correlation between some pairs of biological replicates. This possibility is supported by our observation that almost all BFG-GI strains showing *GIS* $r < 0.5$ were recipients, whereas donors—for which constructs are carried on plasmids—showed *GIS* $r > 0.5$. In the BFG-GI protocol, once the donor and recipient barcodes are fused, the "relic" donor plasmid is counter-selected with 5-FOA to reduce the chance of undesired recombination events. We concur with Jaffe *et al* (2017) who suggest that future protocols using constructs located on plasmids, such as the one we used with the proDonor strains, or at other chromosomal loci could eliminate this issue. Despite this issue, the BFG-GI method proved to be highly accurate when compared with previous benchmark studies.

Although this study focused on a relatively small matrix ($34 \times 38$ genes), we elaborated on previous studies to optimize the two main bottlenecks of pooled cultures: mating (Soellick & Uhrig, 2001) and sporulation (Codon *et al*, 1995). We calculated that to cover a yeast genome-scale matrix of $5,500 \times 5,500$ genes, with 1,000 representative cells for each cross, we would need ~$3 \times 10^{10}$ cells at each step

along the BFG-GI procedure. Furthermore, using the optimal conditions that we established for mating (22%) and sporulation (18%), an experiment covering all $5,500 \times 5,500$ crosses would need to culture pools in ~27 Bioassay 500-cm$^2$ dishes for mating and ~10 l of liquid media for sporulation. Thus, in principle, BFG-GI could be extended to genome-scale studies.

BFG-GI is a flexible technique that can be used in the future to identify genetic interactions in many different settings. Generation of BFG-GI proDonor and proRecipient strains is one of the most time-consuming steps in our pipeline because it includes sequence verification of both *loxP/lox2272* sites and barcodes. However, once generated, these proDonor and proRecipient "toolkits" can be used many times to create donor and recipient strains representing different genes with minimal robotic manipulation. We anticipate that BFG-GI will be a valuable technology to map condition-dependent genetic interactions in yeast and, as next-generation sequencing costs continue to decrease, BFG-GI can be expanded to interrogate pools of double mutants representing bigger sets of gene pairs, including full genome combinations, across multiple conditions.

## Materials and Methods

### Selected DNA repair and neutral gene strains

We retrieved strains representing 26 DNA repair genes whose null mutants were sensitive to MMS (St Onge *et al*, 2007) from the YKO and SGA query collections. Additionally, 14 other deemed-neutral loci were selected, based on lack of evidence that their null mutations affected cell fitness (Table EV2). These 14 loci have few or no genetic interactions in genome-scale screens (Costanzo *et al*, 2010), and we did not find growth defects upon deletion of any of them.

### BFG-GI toolkit strains

#### Donor toolkit construction

We constructed 60 donor strains by generating two DNA fragments with overlapping ends. These were co-transformed into yeast where they recombined to generate pDonor constructs (Fig EV1). The first fragment, called preD1, contained the hygromycin resistance gene (*HygR*) driven by the *Schizosaccharomyces pombe TDH1* promoter and terminator, a barcode locus bearing a 20-bp unique barcode flanked by *loxP/2272* sites, and flanking primer sites. First, we used Gibson assembly (Gibson, 2009) to produce plasmid pFR0032 with the $P_{spTDH1}$-*HygR*-$T_{spTDH1}$ backbone. Then, we used three consecutive PCRs to add barcodes, priming sites, *loxP/2272* loci, and in-yeast recombination adapters (Fig EV1A). The second fragment, preD2, contained the *URA3* marker and Cre recombinase driven by $P_{tetO-CMV}$. We generated this fragment by Gibson assembly of pFR0026, followed by a PCR to add in-yeast recombination adapters (Fig EV1B). Then, preD1 and preD2 fragments were co-transformed into yeast strain RY0771 (derived from BY4742) and merged by in-yeast assembly to generate pDonor plasmids (Fig EV1C). We arrayed transformant strains to extract DNA and sequenced the preD1 loci, and proceeded with those strains containing confirmed preD1 loci. We mated selected *MAT*alpha proDonors with *MAT***a** deletion strains of interest (i.e., DNA repair or neutral genes) from the YKO collection

(Fig EV1D). A series of selective passages (Figs EV1D and EV3) resulted in Donor strains with the relevant genotype:

$MAT$alpha $lyp1\Delta::P_{STE3}$-LEU2 his3$\Delta$1 leu2$\Delta$0 met17$\Delta$0 ura3$\Delta$0 yfg1$\Delta$::kanMX pDonor($P_{tetO\text{-}CMV}$-Cre lox2272 $P_{TDH1}$-HygR-$T_{TDH1}$ barcode loxP $P_{URA3}$-URA3 CEN/ARS $P_{AmpR}$-AmpR ori).

### Recipient toolkit construction

We constructed 56 recipient strains using a method based on the previously described *delitto perfetto* construct (Storici & Resnick, 2006) to enhance homologous recombination of constructs as follows. First, we used consecutive PCRs to produce a fragment preR1, containing the *Kluyveromyces lactis* URA3 gene, flanked by loxP/2272 sites, 20-bp unique barcodes, and a sequence complementary to the *S. cerevisiae* CAN1 locus (Fig EV2A). Second, we incorporated the $P_{STE2}$-spHis5-$T_{STE2}$ into the CAN1 locus of the strain BY4741. Then, the *delitto perfetto* construct was inserted upstream of the *MAT***a** selection reporter of the same strain (Fig EV2B) to enhance homologous recombination of preR1 fragments. This generated a pool of RY0766 proRecipient strains (Fig EV2C). We isolated and arrayed monoclonal proRecipient strains and then sequenced and selected strains with intact preR1 loci. Selected *MAT***a** proRecipients were mated with *MAT*alpha strains of the SGA query collection representing DNA repair and neutral genes (Fig EV1D). A series of selective passages (Figs EV2D and EV3) resulted in recipient strains with the relevant genotype:

*MAT***a** his3$\Delta$1 leu2$\Delta$0 met17$\Delta$0 lyp1$\Delta$ ura3$\Delta$0 can1$\Delta$::barcode loxP klURA3 lox2272 $P_{STE2}$-spHis5-$T_{STE2}$ $P_{CMV}$-rtTA I-SceI $P_{GAL1}$-ISceI yfg2::natMX

## Generation of BFG-GI double mutants

We took several steps to reduce the chance of undesired strains in BFG-GI from taking over pooled cultures. This included optimization of both mating and sporulation, and adapting protocols and molecular constructs that have been reported to improve the selection of the *MAT***a** double-mutant progeny in SGA. Mating and sporulation are the two primary population bottlenecks when generating haploid double mutants by meiotic segregations. As described below, we sought to optimize cultures at these stages to maintain a pool complexity which was large enough to interrogate all desired gene–gene combinations. Optimizing these two processes is also important to reduce potential jackpot effects in pool cultures (i.e., to avoid strains with genetic anomalies to take over the entire pool growth).

### Mating optimization for en masse BFG-GI

We focused on optimization of cell density for *en masse* party mating because previous evidence shows cell density influences mating efficiency (Soellick & Uhrig, 2001). We determined the optimal cell density for *en masse* party mating by inoculating mating Petri dishes with a mixture of two neutral strains (*MAT*alpha Donor ho$\Delta$:: kanMX, and *MAT***a** Recipient ylr179c$\Delta$::natMX) at cell densities varying from $3 \times 10^8$ to $3 \times 10^9$ per dish. After generating mating mixtures, we took samples at 0 and 12 h of incubation at 23°C, and inoculated plates with either non-selective or heterozygous diploid double-mutant selective media and counted colony-forming units (CFUs). The ratio of CFUs in non-selective versus selective media indicated that inoculating a 58-cm$^2$ Petri dish with $3 \times 10^8$ cells of

mating mixture resulted in 22% mating efficiency. In contrast, $1 \times 10^9$ cells of mating mixture resulted in 13% mating efficiency, and $3 \times 10^9$ cells of mating mixture resulted in 3% mating efficiency. Hence, we used $5.1 \times 10^6$ cells of mating mixture per cm$^2$ of plate for further *en masse* party matings.

To generate pools of double mutants, we arrayed BFG-GI donors and recipients in their respective selective media and cultured at 30°C for 48 h (Fig EV3). We made one pool for each mating type by scraping cells from plates into liquid media and normalized cell densities with 1 M sorbitol to have equal number of cells per strain ($5 \times 10^8$ cells per ml) for each pool. Then, we lightly sonicated cells to disrupt clumps (Branson microtip sonicator, 10% duty cycle, output 2, 25 bursts, pause of 3 s, and a second 25 burst). We mixed the two pools together by stirring them in a flask for 10 min. Finally, we inoculated two Bioassay dishes (500 cm$^2$) with $2.59 \times 10^9$ cells each of the mating mixture, and mating cultures were incubated for 12 h at 23°C (Fig EV3).

### Generation of heterozygous diploid double mutants, induction of barcode fusion, and pDonor elimination

Generation of heterozygous diploid double mutants required passaging the mating progeny every 24 h into fresh selective media. Passages included selection of heterozygous diploid double mutants, induction of the Cre/Lox system with doxycycline, counter-selection of the relic pDonor with 5-FOA, and recovery from 5-FOA counter-selection to increase sporulation efficiency (Fig EV3).

### Sporulation optimization for en masse BFG-GI

We used cultures recovered from 5-FOA counter-selection to inoculate liquid PRE5 pre-sporulation media (Codon *et al*, 1995) for 2 h at 30°C to induce exponential growth, then spun down the cells, and transferred them to SPO2 sporulation media (Codon *et al*, 1995) supplemented with histidine, leucine, methionine, and uracil to mask BFG-GI strain auxotrophies at concentrations used in the SGA sporulation protocol (Tong & Boone, 2005). We incubated sporulation cultures at 21°C for 12 days. This resulted in ~18% sporulation efficiency, as evaluated by counting CFUs in non-selective and selective media and tetrad visualization. Shorter incubation periods reduced the sporulation efficiency (~4% at 5 days, ~13% at 7 days).

### Selection of MATa haploid double mutants with fused barcodes

We selected *MAT***a** haploid progeny from sporulation cultures, followed by haploid double-mutant selection (Fig EV3). Aliquots were stored in glycerol at −80 degrees for future use. We used the STE2 and STE3 promoters currently used for SGA to select for haploid cells, as markers with these promoters have been reported to perform better than earlier alternatives (e.g., *MFA1/MFA2* promoters) (Tong & Boone, 2007). We used these constructs to first select the *MAT***a** progeny from sporulation cultures and then the haploid double mutants. Using *STE2/STE3* promoters, optimizing mating and sporulation, and using an intermediate *MAT***a** selection step between sporulation and haploid double-mutant selection together likely reduced the number of mitotic crossover survivors and jackpot mutation effects in our pools.

### Exposure of pooled cultures to drugs

Before challenging haploid double-mutant pools to drugs, we identified the appropriate drug concentration for our experiment by

exposing a neutral BFG-GI haploid double mutant (*hoΔ::kanMX/ylr179cΔ::natMX*) in growth assay liquid cultures to various drug concentrations. We selected drug doses corresponding to 20% of the minimal inhibitory concentration for the neutral test strain (Table EV1). To expose mutant strains to drugs, we thawed frozen haploid double-mutant pools, allowed the pools to recover for 2 h in haploid double-mutant liquid media at 30°C, and then used $1 \times 10^9$ cells of this culture to inoculate 143-cm$^2$ Petri dishes containing solid media supplemented with each DNA repair drug. We cultured pools at 30°C for 24 h and then collected samples to sequence fused barcodes and thus infer the abundance of each double-mutant.

### Generation of BFG-GI double mutants in an array format

Mating and selecting donor and recipient strains in an array format was similar to the pool-based *en masse* party mating assay described above, but in this case, we used robotic assistance to pairwise mate each donor with an array of recipients. We completed all steps, including sporulation, on solid media, and imaged the final haploid double-mutant selection plates. We scraped cells from the final selection plates to sequence the fused-barcode population which allowed us to compare cell patch sizes with numbers of sequencing reads.

### Next-generation sequencing and mapping of fused barcode pairs

The BFG-GI technology relies on the Cre/Lox system to recombine the complementary donor and recipient *loxP/lox2272* sites that serve to introduce the donor barcode adjacent to the recipient barcode (Fig 1). We multiplex-sequenced the fused barcodes from pools of cells using the following steps: (i) genomic DNA extraction using glass beads and phenol/chloroform; (ii) PCR amplification of the 325-bp barcode fusion product including the two 20-bp barcodes and the multiplexing sequencing adapters (one index for each condition, for each technical replicate); (iii) concentration and gel purification of amplicons using 2% E-Gel EX agarose 2% (Invitrogen), DNA Clean & Concentrator Kit (Zymo Research), and MinElute Gel Extraction Kit 50 (Qiagen); (iv) normalization of DNA libraries using Qubit Fluorometric Quantitation (Invitrogen); (v) mix of libraries at equal concentrations; (vi) quantification of the pooled DNA library mix by qPCR; and (vii) sequencing by Illumina 75-cycle NextSeq paired-end technology, including 25 cycles for each barcode and 6 cycles for the multiplex index. We mapped sequencing *.fastq files against the library of expected barcode sequences using the program Segemehl (v0.1.7, -A 85) and custom scripts; 97% of all sequencing reads mapped to expected barcodes.

### Whole-genome sequencing and detection of chromosome duplications

Ten strain pairs with one strain with *GIS* r < 0.5 and another with *GIS* r > 0.5 with other replicates for the same gene were selected for genome sequencing. Genomic DNA from 20 strains was extracted via cell wall disruption with Zymolyase 100T 10 mg/ml (Amsbio) and purification using AMPure beads (Agilent). gDNA was quantified with Quant-iT PicoGreen dsDNA assay kit (Invitrogen) and normalized to 2 ng/μl for DNA fragmentation and library normalization with a Nextera XT DNA Library Prep Kit, using a transposase (Tn5) for tagmentation. A limited-cycle PCR was used to add Illumina sequencing adapters and indices i5 and i7. PCR amplicons with size between 400 and 800 bp were gel-purified using a 2% E-Gel EX agarose 2% (Invitrogen) and MinElute Gel Extraction kit

(Qiagen). Whole-genome sequencing was conducted on an Illumina NextSeq 500 using a HighOutput 150 cycles v2 kit with 40× coverage. Sequencing results were mapped against the reference genome UCSC sacCer3 (SGD vR64.1.1), corrected for GC content, and chromosomal duplications detected with the HMMcopy R package (Ha *et al*, 2012).

### Retesting double-mutant construction and spot dilution assays

We generated double-mutant strains for retesting in spot dilution assays by mating single-mutant *MAT*alpha SGA queries with *MAT***a** YKO collection strains, the exceptions being the *MAT***a** *RAD53* (MBS1437) and *rad53-R605A* (MBS1440) strains with the *RAD53* loci linked C-terminally to a *6xHis-3xFLAG-kanMX6* tag and resistance marker (Ohouo *et al*, 2013). Next, we induced sporulation of heterozygous diploid double mutants as we did for BFG-GI strains. To confirm segregation of *kanMX* and *natMX* markers, we manually dissected haploid double mutants from tetrads and verified segregation using both selective media and PCR. Sanger sequencing confirmed the proper identity of residue 605 in intact *RAD53* and *rad53-R605A* strains. We grew strains overnight to saturation in liquid media, diluted them 1:10, and then used 1:5 serial dilutions for the spot assays. All cultures used YPD media supplemented with indicated drug concentrations.

### Defining a genetic interaction score (*GIS*)

In an exponential growth model, the frequency of a double-mutant strain $s_{xy}$ in a given condition at a time $t$ ($f_{s_{xy},t}$) represents its total growth from an initial number $f_{s_{xy},t=0}$ as a proportion of the total growth of all other strains in the pool:

$$f_{s_{xy},t} = \frac{N_{s_{xy},t=0}2^{g_{xy}t}}{\sum N_{s_{ij},t=0}2^{g_{ij}t}}$$

Note: Before calculating frequency, we add a pseudocount of 0.5 to the count of every strain in our analysis to avoid a zero denominator in several calculations.

Here, $g_{xy}$ is inversely related to the doubling time of strain $s_{xy}$ and $g_{xy}t$ effectively represents the number of doublings of strain $s_{xy}$. Units for $t$ can be chosen arbitrarily. In this model, a frequency at $t = 0$ evaluates as:

$$f_{s_{xy},0} = \frac{N_{s_{xy},t=0}}{\sum N_{s_{ij},t=0}}$$

To remove the unknown $N_{s_{ij},t=0}$ term, we define $r_{s_{ij},t}$:

$$r_{s_{xy},t} \equiv \frac{f_{s_{xy},t}}{f_{s_{xy},0}} = 2^{g_{xy}t}\frac{\sum N_{s_{ij},t=0}}{\sum N_{s_{ij},t=0}2^{g_{ij}t}}$$

We note that the $\frac{\sum N_{s_{ij},t=0}}{\sum N_{s_{ij},t=0}2^{g_{ij}t}}$ term is the ratio between the initial and final number of cells in the pool and can be calculated by the total number of generations of pool growth ($gen_{pool}$):

$$\frac{\sum N_{s_{ij},t=0}}{\sum N_{s_{ij},t=0}2^{g_{ij}t}} = \frac{1}{2^{gen_{pool}}}$$

Therefore, $g_{xy}t$ can be calculated as:

$$g_{xy}t = \log_2(r_{s_{xy},t}) + gen_{pool}$$

To calculate $g_{wt}t$, we take the mean $g_{xy}t$ of all neutral-neutral pairs:

$$g_{wt}t = \text{mean}(g_{ij}t \mid ij \in \text{neutral})$$

We then obtain the relative growth rate $w_{xy}$ of each strain compared to the wild type by dividing their number of doublings. In a constant exponential growth model, this metric is independent of time. In practice, $g$ represents the average growth rate over the measured time period.

$$w_{xy} = \frac{g_{xy}t}{g_{wt}t} = \frac{g_{xy}}{g_{wt}}$$

To estimate the single-mutant fitness $w_x$ and $w_y$ for a given pair, we use the mean estimate of $x$ or $y$ combined with neutral genes.

$$w_x = \text{mean}(w_{xj} \mid j \in \text{neutral})$$

$$w_y = \text{mean}(w_{iy} \mid i \in \text{neutral})$$

We then define the genetic interaction score (*GIS*) as the difference between $w_{xy}$ and the product of $w_x$ with $w_y$:

$$GIS_{xy} \equiv w_{xy} - w_x w_y$$

Because there is uncertainty in $w$, it is possible to calculate $w < 0$ for $w_x$, $w_y$, or $w_{xy}$. Such values are assigned as 0 when performing the *GIS* calculation.

### Normalizing genetic interactions and calculating *P*-values

To assign a threshold for positive and negative genetic interactions, several additional steps are performed. $GIS_{xy}$ is converted to a standard score by calculating how many standard deviations $GIS_{xy}$ is from 0 given an estimate of $GIS_{xy}$ uncertainty ($\hat{\sigma}_{GIS_{xy}}$).

$$Z_{GIS_{xy}} = \frac{GIS_{xy}}{\hat{\sigma}_{GIS_{xy}}}$$

To calculate $\hat{\sigma}_{GIS_{xy}}$, we identify various sources of uncertainty. Another way to state $GIS_{xy}$ is as such:

$$GIS_{xy} = w_{xy}t - \frac{g_x t g_y t}{g_{wt}t}$$

We then define an error model to calculate the standard error $\sigma$ for each term used in this calculation:

$\hat{\sigma}_{w_{xy}t}$: This is estimated globally for each condition as the median difference between $w_{xy}t$ between the R1 and R2 technical replicates for all strains. We note that this error model only captures the general expected error between two separate runs of the same biological sample.

$\hat{\sigma}_{g_x t}$, $\hat{\sigma}_{g_y t}$, $\hat{\sigma}_{g_{wt}t}$: Each of these $g$ values is calculated by taking the mean of multiple strains. We use the variation of growth estimates in these strains (i.e., the standard deviation) as the uncertainty.

The delta method for approximating the propagation of measurement uncertainty is used to combine $\hat{\sigma}_{w_{xy}t}$, $\hat{\sigma}_{g_x t}$, $\hat{\sigma}_{g_y t}$ and $\hat{\sigma}_{g_{wt}t}$ into $\hat{\sigma}_{GIS_{xy}}$. This formula is also used for obtaining the other error estimates reported (i.e., $\hat{\sigma}_{w_x}$, $\hat{\sigma}_{w_y}$, $\hat{\sigma}_{w_{xy}}$).

To assign a *P*-value for each interaction, we then analyze the distribution of $Z_{GIS}$ in all unlinked neutral-neutral and neutral-DNA damage pairs (hereafter called "neutral pairs"), as few or no genetic interactions are expected to take place in this space. We model $Z_{GIS_{np}}$ as a normal distribution (Fig EV4D shows the empirical and fitted normal distribution for each condition to validate this decision) and use the *pnorm* function in R to calculate $p_{pos} = p(Z_{GIS_{neutral}} \geq Z_{GIS_{xy}})$ and $p_{neg} = p(Z_{GIS_{neutral}} \leq Z_{GIS_{xy}})$ for each pair. We then combine these single-tailed tests into a two-tailed value:

$$p_{neutral} = \min(p_{pos}, p_{neg}) \times 2$$

$p_{neutral}$ represents the probability that a score as extreme as $Z_{GIS_{xy}}$ or more would be found amongst neutral pairs.

### Combining multiple biological replicates and calculating a FDR

We consolidated multiple measurements of $w_x$, $w_y$, $w_{xy}$, $GIS_{xy}$, $Z_{GIS_{xy}}$ as well as $\hat{\sigma}_{w_x}$, $\hat{\sigma}_{w_y}$, $\hat{\sigma}_{w_{xy}}$, $\hat{\sigma}_{GIS_{xy}}$ and $p_{neutral}$ from multiple barcode pairs into a single value for each gene pair. $GIS_{xy}$ values were weighted by the inverse of estimated squared error ($w = \frac{1}{\hat{\sigma}_{GIS_{xy}}^2}$) and averaged to obtain $GIS_{gene_x,gene_y}$. Similarly, $w_x$, $w_y$, $w_{xy}$ were averaged by the same weight ($w$) to obtain their corresponding gene-wise value. $\hat{\sigma}_{w_{gene_x}}$, $\hat{\sigma}_{w_{gene_y}}$, $\hat{\sigma}_{w_{gene_x,gene_y}}$, $\hat{\sigma}_{GIS_{gene_x,gene_y}}$ were obtained using the propagation of uncertainty when calculating a weighted average:

$$\hat{\sigma}_{gene} = \sqrt{\sum \hat{\sigma}_{barcode}^2 \left(\frac{w}{\sum w}\right)^2}$$

$Z_{GIS_{gene_x,gene_y}}$ was calculated using $GIS_{gene_x,gene_y}$ and $\hat{\sigma}_{GIS_{gene_x,gene_y}}$:

$$Z_{GIS_{gene_x,gene_y}} = \frac{GIS_{gene_x,gene_y}}{\hat{\sigma}_{GIS_{gene_x,gene_y}}}$$

Finally, a gene-wise $p_{neutral}$ was calculated using Stouffer's method weighted by $w$. The gene-wise $p_{neutral}$ values were then converted to $FDR_{neutral}$ using the *qvalue* function in the *qvalue* R package.

### Calling differential genetic interactions

For each gene pair, we calculated $\Delta GIS$ and $\Delta Z$ for all pairwise comparisons ($a$–$b$) amongst the tested conditions. $\Delta GIS_{gene_x,gene_y,a-b}$ was calculated as $GIS_{gene_x,gene_y,a} - GIS_{gene_x,gene_y,b}$, and $\Delta Z_{GIS_{gene_x,gene_y,a-b}}$ was calculated as:

$$\Delta Z = \frac{GIS_{gene_x,gene_y,a} - GIS_{gene_x,gene_y,b}}{\sqrt{\hat{\sigma}_{GIS_{gene_x,gene_y,a}}^2 + \hat{\sigma}_{GIS_{gene_x,gene_y,b}}^2}}$$

For each pair of conditions, $\Delta Z$ was calculated for all unlinked neutral-neutral and neutral-DNA damage pairs ("neutral pairs") to create a null distribution for $\Delta Z_{neutral}$. $p_{\Delta neutral}$ was then calculated for each pair from the $\Delta Z_{neutral}$ distribution in the same manner as calculating $p_{neutral}$. $p_{\Delta neutral}$ values were then converted to $FDR_{\Delta neutral}$ using the *qvalue* function in the *qvalue* R package.

### Data availability

Raw and normalized sequencing measurements and GIS for each gene pair are available in Tables EV2–EV6, and Code EV1, written in R (R Core Team, 2017), allows to generate Tables EV3–EV6 from Table EV2. Any modifications post-publication will have been documented at https://github.com/a3cel2/BFG_GI_stats.

**Expanded View** for this article is available online.

### Acknowledgements

We are grateful for helpful comments from Yong Lu, Michael Principato, Ramamurthy Mani, and Meng Xiao He at the outset of this project, to Brenda Andrews and Charles Boone and members of their laboratories for providing reagents and insightful comments, and to many Roth Lab members for support and feedback throughout this project. We gratefully acknowledge support by the Canadian Excellence Research Chairs (CERC) Program (to FPR), Canadian Institutes of Health Research (MOP-79368 to GWB, and FDN143343 to DD), National Human Genome Research Institute of the National Institutes of Health (NIH/NHGRI) HG004756, and by an individual NRSA award (HG004825) to JCM. FPR was also supported by a NIH/NHGRI Center of Excellence in Genomic Science (HG004233), by NIH/NHGRI Grant HG001715, and by the One Brave Idea Foundation.

### Author contributions

FPR, JJD-M, JCM, and ACo (Atina Coté) conceived the project; ACo, AK, and SO constructed pilot strains. ACo, PB, CW, and JR constructed final BFG-GI strains and performed pilot mating experiments. JJDM, FS, YZ, DAP, and GG optimized mating, marker selection, and sporulation protocols. JJD-M, FS, ACo, MG, and MV performed array and pool cultures and sequencing of fused barcodes. JJD-M and JCM performed computational mapping of barcodes. JJD-M and ACe (Albi Celaj) performed scoring of genetic interactions. JJD-M and MG performed aneuploidy experiments, and JW analyzed the results. JJD-M, AB, and BH performed Shu complex-related experiments. DD and GWB provided advice on DNA repair pathways. JJD-M, FPR, GWB, AB, and ACe wrote the manuscript. FPR supervised the project.

### Conflict of interest

The authors declare that they have no conflict of interest.

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
