## [Review Process File · Molecular Systems Biology]

Mapping DNA damage-dependent genetic interactions in yeast via party mating and barcode fusion genetics

J. Javier Díaz-Mejía, Albi Celaj, Joseph C. Mellor, Atina Coté, Attila Balint, Brandon Ho, Pritpal Bansal, Fatemeh Shaeri, Marinella Gebbia, Jochen Weile, Marta Verby, Anna Karkhanina, YiFan Zhang, Cassandra Wong, Justin Rich, D'Arcy Prendergast, Gaurav Gupta, Sedide Öztürk, Daniel Durocher, Grant W. Brown & Frederick P. Roth

Review timeline:

Submission date:	7 September 2017
Editorial Decision:	29 September 2017
Revision received:	8 April 2018
Editorial Decision:	23 April 2018
Revision received:	30 April 2018
Accepted:	2 May 2018

Editor: Maria Polychronidou

Transaction Report:

1st Editorial Decision

29 September 2017

Thank you again for submitting your work to Molecular Systems Biology. We have now heard back from the three referees who agreed to evaluate your study. As you will see below, the reviewers are overall quite positive and think that the proposed approach seems useful. They raise however a series of concerns, which we would ask you to address in a revision of the manuscript.

The reviewers' recommendations are rather clear and mostly refer to relatively minor issues. Therefore, I think that it is not required to repeat all the points listed below. Related to the follow-up experiments on the Shu complex, reviewer #2 suggests some additional analyses that would allow better supporting the conclusion on the regulation of Rad53 by the Shu complex.

REVIEWER REPORTS

Reviewer #1:

The authors describe an approach for quantifying genetic interactions between yeast gene deletions using a pooled assay using deep sequencing. They illustrate how this can be useful by quantifying a panel of potential interactions between DNA repair genes across multiple environmental conditions (in the presence of different DNA damaging agents). In general the approach should be useful, the manuscript is well written, and the results nicely exemplify how this can be used to gain biological insight.

Comments

The calling of differential genetic interactions between conditions needs to be improved. At the moment, if I understand correctly, they use a Z-score cut-off in each condition to define interactions in that condition and then they simply count whether an interaction was found or not in each condition. This is likely to call false positive differential interactions because of fluctuations around the Z-score threshold. I would like to see quantitative comparisons of the interaction strengths across the conditions and direct statistical tests for the differences in interaction between / across the conditions and of course to control the global false discovery rate.

The method is conceptually and practically related to approaches developed by the authors and other groups for the parallel analysis of protein-protein interactions. It would be good to highlight this in the introduction.

It would be sensible to avoid the term 'orgy mating'.

Reviewer #2:

In the manuscript "Mapping DNA damage-dependent genetic interactions in yeast via orgy mating and barcode fusion genetics" the authors created a new system to assay double mutant knockouts for fitness upon different experimental conditions using a unique barcoding system. This system has advantages of being able to freeze the parental pool and then to expose this pool to different experimental conditions at a later date. The analysis shown indicated a very good correlation with the traditional SGA methodology comparing sequencing reads to colony size ($R = 0.92$). Using this approach, the authors discovered a novel genetic interaction between the Shu complex with SLX4, PPH3, and RAD53. Overall the manuscript represents advancement in SGA screening and provided some new genetic interactions with the Shu complex.

Major comments:

- 1) It's unclear how many proRecipient strains (or proDonor strains) were created using this new method and what representation it is of the genome.
- 2) The list of the genes queried in Supplemental Figure 4 should be indicated in a table or in the figure. It would also be nice to have the results of the screen available since those would be of interest to the DNA repair community (Figure 3 is very hard to see and so supplemental spreadsheets with the data would be very helpful- it could also be color coded to correlate with Figure 3C- orange, pink, tan, green, red and blue).
- 3) As it stands, there is not enough direct evidence provided to support the model that the Shu complex regulates Rad53. The language should be softened to "may regulate" or "leads to a decrease in Rad53 activity/activation" instead. Alternatively, more experimental evidence should be provided. An alternative hypothesis is that if the Shu complex is disrupted and the cells are exposed to MMS, Rad53 is hyper-activated because the Shu complex isn't present to repair the damage. If Rad53 signaling is inhibited by the R605A mutation, then the Shu complex may not be needed because of the inability for the cell to detect the damage or perhaps the use of an alternative repair pathway to bypass the lesion (such as TLS).
- 4) The serial dilutions presented in Figure 4E and 4G are difficult to compare because of the differences observed between Shu complex mutants. In most cases where the Shu complex has been analyzed in vivo, individual Shu complex disruptions (SHU1, SHU2, CSM2, PSY3) leads to the same phenotypes. Occasionally SHU1-SHU2 and PSY3-CSM2 will have different results (only observed in meiosis) but they are consistent with their binding partners.
 - a. In Figure 4E, *shu1Δ pph3Δ* cells are less sensitive than the other Shu complex members. It is possible that these differences have to do with incubation times or having results from different

plates/controls. The serial dilutions need to be redone on a single plate for a more accurate comparison.

b. In Figure 4G, differences are seen this time with *rad53-R605A shu2Δ*. Since disruption of the Shu genes leads to increased mutation rates (due to TLS), it's possible that this double mutant has a suppressor and that is why there is better growth, which is comparable to a *rad53-R605A* single mutant. Again, because the suppression is difficult to observe except for the case of *rad53-R605A shu2Δ* (which looks different from the other Shu complex members), it important to redo the serial dilutions on the same plate or a few plates (with paired controls such as WT, *rad53-R605A*, *shu1Δ*, *rad53-R605A shu1Δ*, *shu2Δ*, *rad53Δ-R605A*, etc...).

Minor comments:

1) The sentence in the Discussion, p. 19 "Thus, our results provide evidence for a previously uncharacterized role of the Shu complex in the cellular response to DNA damage by MMS." is inaccurate statement. The Shu complex role has primarily been characterized upon MMS damage but the novel part provided is the link to Rad53.

2) On p.17 the authors state "SLX4 shows epistatic relationship with error-free lesion bypass genes during MMS treatment (Flott et al 2007)". Is the same phenotype observed in this study?

3) On p. 13 Please mention the "10 culture conditions" in the main body or refer to Fig 3C for clarity.

Reviewer #3:

Díaz-Medjía et al. introduce a new technique to generate yeast double mutants and subsequently assay them across multiple conditions. With respect to the numerous techniques already available, the proposed approach has the advantage to allow the generation of a single pool containing virtually all the desired double mutants. This means that such pool could be easily tested across many conditions with minimal labour and therefore with high-throughput. The authors tested the approach in a 34x38 gene combinations, and across 10 conditions related to DNA damage. They demonstrated how the technique could recapitulate previous experiments and genetic interaction screens, and went to further inspect a newly uncovered genetic interaction (SLX4 and the Shu complex). The term "orgy mating" is surely catchy, and also the probable reference to a popular 90s videogame (BFG) has been noted.

The technique is sufficiently detailed so that the minute details can be easily inspected. Furthermore the authors included the original read counts and final genetic interaction score (GIS), allowing in-depth inspection of their proposed scoring scheme. Numerous other techniques are already available, including one that is conceptually very similar despite being more reliant on robotic work (iSeq, Jaffe et al., 2017). The authors fully acknowledge the existence of all these alternatives, and introduce compelling arguments why their own implementation can potentially streamline genetic-interaction screening across multiple conditions. The authors also acknowledge how their method shares the same weakness as iSeq, namely the development of aneuploidies when using the *can1* locus in generating the recipients, and they indicate how this could be fixed in future developments of the technique.

There are a number of issues with this manuscript, although none particularly serious. I wasn't able to reproduce the GIS from the read counts provided, even if to the best of my knowledge I have implemented the author's same GIS scoring scheme. It seems that the authors have carried out some form of quantile normalization to their raw GIS, despite not reporting it in the method's section. After looking at supplementary figure S4 I think the problem might be related to the partition of the tested genes into "neutral" and "related to DNA damage", even though how this separation comes into play into the GIS computation is evident only for the wild-type fitness calculation. This discrepancy also didn't allow to verify whether an absolute z-score cutoff of 1 is appropriate. The authors could provide a more detailed description of their scoring scheme or ideally a code implementation of their scoring procedure.

Regarding the reproducibility of the approach: the authors have decided to merge the two technical replicates after noticing that the "relative strain abundance" (I'm assuming equivalent to Fijk as reported in Materials and methods) correlation was above 0.95. I would be interesting in knowing as well the correlation in GIS. Read counts are bound to be highly correlated, so I'm not sure how that measure is really informative with respect to reproducibility.

Related to the previous point is also the choice of plotting the (raw?) colony size against read counts as a measure of comparability between "conventional" SGA and the proposed new method (figure 2A). Wouldn't a plot comparing GIS and something like an S-score be more appropriate?

The authors have compared their approach to a previous study (St Onge et al., 2017) in two conditions. The scatter plots comparing the epsilon score with the GIS seem to indicate that the two scoring schemes have very different dynamic ranges; this is particularly evident when looking at the "no drug" condition. Can the authors comment on these differences?

In previous studies of conditional genetic interactions (example: Bandyopadhyay et al, 2010) it was noted that the variance of the difference between genetic interaction scores depends on the magnitude of this difference. In other words, the most extreme genetic interaction scores tend to have higher variance across replicates than lower values. In these studies the difference in genetic interactions was therefore taking into account this dependency between variance across replicate and magnitude in difference in genetic interaction scores. This resulted in a somewhat non-intuitive set of conditional genetic interactions across conditions that was defined as a neutral genetic interaction in both condition. Would this also apply to the genetic interaction scores as defined in this study ?

In figure 2F the authors use same gene pairs and linked genes to show how those gene pairs GIS are enriched in negative values. Why is the "same gene pairs" curve showing a shoulder above zero?

The authors report that the double barcode total size is 325bp and how they have used an Illumina NextSeq PE platform. What was the read length? Is it sufficient to reliably distinguish pairs of barcodes? Would that be true if the number of tested genes was way higher?

To what extent can such this method to scale up, eventually to a genome-wide level ? The authors report how they optimised their mating setup for their 34x38 gene matrix, but it would perhaps be appropriate if they could add even just a theoretical account on the upper limit in terms of gene pairs that can be generated with their approach.

Minor comments:

There is a section of discussion (paragraph 6) that seems could be moved either in the results or the materials and methods section, or at least part of it.

In figure 3F a gene (RAD57) can be seen switching the direction of its genetic interaction quite often (in 13 condition pairs). Is this an unexpected result?

How many of the replicates with GIS $R < 0.5$ are recipients? An absolute number would be more accurate.

1st Revision - authors' response

8 April 2018

We appreciate the thorough, thoughtful and constructive comments and suggestions from you and the reviewers.

In response, we have made major improvements to our analysis methodology. This had an impact on the resulting genetic interaction map, increasing the number of positive interactions and also agreement of our map with a previous genetic interaction map. We have also carried out more experiments related to Rad53 and the Shu complex.

We hope that you will agree that all of the issues raised are addressed by our responses below and by changes to the manuscript.

In our response below, editorial and review comments are shown in *italics*, with comment numbering in red and our responses in blue text.

Reviewer #1:

The authors describe an approach for quantifying genetic interactions between yeast gene deletions using a pooled assay using deep sequencing. They illustrate how this can be useful by quantifying a panel of potential interactions between DNA repair genes across multiple environmental conditions (in the presence of different DNA damaging agents). In general the approach should be useful, the manuscript is well written, and the results nicely exemplify how this can be used to gain biological insight.

We appreciate the thoughtful and constructive comments.

Comments

R1-C1

The calling of differential genetic interactions between conditions needs to be improved. At the moment, if I understand correctly, they use a Z-score cut-off in each condition to define interactions in that condition and then they simply count whether an interaction was found or not in each condition. This is likely to call false positive differential interactions because of fluctuations around the Z-score threshold. I would like to see quantitative comparisons of the interaction strengths across the conditions and direct statistical tests for the differences in interaction between / across the conditions and of course to control the global false discovery rate.

With hindsight, we agree that our procedures for calling both interactions and differential interactions were somewhat *ad hoc*. In response to this comment and others, we brought in another author (ACe), who has substantially improved our statistical analysis.

Here we describe changes to the procedure for calculating Genetic Interaction Score (GIS), describe an updated procedure for setting thresholds to call interactions, and finally describe the new procedure for calling differential interactions (which addresses the specific concern raised above).

We developed a new approach to calculating GIS. The new method, which incorporates empirical measurements of the number of doublings of each pool at each time point and uses these to fit an exponential growth model for each strain within each pool, is detailed in the Materials and Methods section (Page 36) of our current submission. This is a theoretical improvement over our previous GIS method, which did not estimate exponential growth rates. We now report these estimated rates for each pair measured.

We have some evidence that this theoretical improvement is also an improvement in practice: the new scoring approach yielded an interaction map for which empirical agreement with the epsilon values reported by St Onge et al (2007) was improved. For example, our previous GIS vs. St Onge Epsilon comparison yielded a correlation of $r = 0.57$ for the “No MMS” condition and $r = 0.75$ for “MMS”; whereas our new GIS yielded $r = 0.8$ for No MMS and $r = 0.85$ for MMS. Receiver-operating characteristic curve (ROC) analysis for the previous and new maps was basically the same for negative genetic interactions, with area under the ROC (AUC) = 0.87 before and 0.89 now. However, positive genetic interactions in the new map showed an even greater improvement (previous AUC=0.8 vs. 0.9 now), supporting the theoretical arguments for the new GIS scoring strategy.

We have also incorporated estimates of error in GIS scores (propagated from estimates of error in the component fitness values that are used as inputs). A new approach for standardizing the GIS score to yield Z-scores was applied. This approach used estimated uncertainty in each GIS calculation, as opposed to the previous Z-score which standardized more naively using the spread of the overall distribution of GIS scores (Page 36).

We also now determine Z-score thresholds such that the false discovery rate (FDR) is kept below 1%. False discovery rate estimation requires a null distribution, and the set of strains in our screen that carry deletions in neutral genes enabled this analysis. Specifically, the distribution of Z-scores for all unlinked gene pairs that involve at least one neutral gene ('neutral pairs') served as the null distribution for estimation of false discovery rates.

Finally, to specifically address this Reviewer's concerns, we carried out FDR analysis for differential interactions. Using the GIS and error models for each gene pair, we can calculate a standardized difference score (ΔZ). The ΔZ distribution amongst neutral pairs served as an empirical null distribution for ΔZ , that was used to estimate and control FDR below 1%. We have further added a differential effect size threshold ($|\Delta GIS| > 0.1$) for these calls to filter out otherwise differential interactions with small effect sizes. The new method is detailed in the Materials and Methods section (Page 40) of our current submission.

This update in methodology has changed specific results in Fig. 2 B-E and Fig. 3. We have updated the relevant numbers in the main text. However, overall our findings remain the same - for example, most condition-dependent genetic interactions were still transitions from neutrality in one condition to a positive or negative interaction in the other (rather than from negative to positive or vice-versa). Notably, Fig. 4B remains completely unchanged, except for newly added confirming experiments. To better describe the new interaction calling procedure and empirically justify our cutoffs, we have added 4 extra panels to Fig. EV4. To better describe and highlight the new differential-interaction calling procedure and we have added an extra figure (Fig. EV5). Furthermore, we note the interesting differential interaction patterns of *RAD5* (Page 18, Fig. EV5 C-D)

R1-C2

The method is conceptually and practically related to approaches developed by the authors and other groups for the parallel analysis of protein-protein interactions. It would be good to highlight this in the introduction.

We are grateful to the reviewer for pointing out this important omission. The revised manuscript now incorporates references to Hastie and Pruitt (*Nucleic Acids Research* 2007), Yachie *et al* (*Molecular Systems Biology* 2016), and Schlecht *et al* (*Nature Communications* 2017), all of which use related methods for protein interaction mapping (Page 7).

R1-C3

It would be sensible to avoid the term 'orgy mating'.

Although we find "orgy mating" to be both clear and evocative, we accept that it may be too racy for some readers. We have changed the title to :

Mapping DNA damage-dependent genetic interactions in yeast via party mating and barcode fusion genetics

Although the term "party" may seem overly casual, we note that the terms "party hub" and "date hub" are widely used and understood within the field of biological network analysis due to the influence of the Han *et al.* (*Nature* 2004) paper which has been cited over 1,500 times. Although we would be willing to replace "party mating" with the drier and more obscure scientific term "panmixia," we would much prefer either "orgy mating" or "party mating", as each of these terms seem accurate while being both transparent and evocative.

Reviewer #2:

*In the manuscript "Mapping DNA damage-dependent genetic interactions in yeast via orgy mating and barcode fusion genetics" the authors created a new system to assay double mutant knockouts for fitness upon different experimental conditions using a unique barcoding system. This system has advantages of being able to freeze the parental pool and then to expose this pool to different experimental conditions at a later date. The analysis shown indicated a very good correlation with the traditional SGA methodology comparing sequencing reads to colony size ($R = 0.92$). Using this approach, the authors discovered a novel genetic interaction between the Shu complex with *SLX4*,*

PPH3, and RAD53. Overall the manuscript represents advancement in SGA screening and provided some new genetic interactions with the Shu complex.

We appreciate the thorough and constructive comments.

Major comments:

R2-C1

1) It's unclear how many proRecipient strains (or proDonor strains) were created using this new method and what representation it is of the genome.

We initially generated 65 proDonor and 71 proRecipient strains, representing 26 DNA repair genes and 14 Neutral loci (described in the current version of the manuscript on Page 12). After quality controls, our final dataset contained 59 Donors and 56 Recipients (Page 14 and Figure EV4). Although our experiments examined fewer than <1% (40 out of ~6000 genes) of protein-coding yeast genes, we note that the key goal of this paper was to describe a scalable method applied at proof-of-principle scale under multiple growth environments, leaving application at genome scale to future studies.

R2-C2

2) The list of the genes queried in Supplemental Figure 4 should be indicated in a table or in the figure. It would also be nice to have the results of the screen available since those would be of interest to the DNA repair community (Figure 3 is very hard to see and so supplemental spreadsheets with the data would be very helpful- it could also be color coded to correlate with Figure 3C- orange, pink, tan, green, red and blue).

Table S2 in the initial manuscript did provide read counts and processed genetic interaction scores for each gene pair; however, we also now provide additional information in four Extended View tables. Table EV2 can be computationally queried by the reader to obtain the full list of genes included in this study.

Regarding data in Figure 3. The new Table EV4 contains columns labeled 'Z_GIS_xy.*_Class' with a label Positive, Negative or Expected for each gene pair on each condition. This data corresponds to the networks in the diagonal of Fig 3C. Likewise, the new Table EV6 contains columns labeled 'Class_Condition1' and 'Class_Condition2' which are the underlying data for off-diagonal networks and barplots in the same Fig 3C. We provide these labels to facilitate the reader to identify/sort genetic interaction type changes. References of EV Tables underlying Fig 3 panels are indicated on the figure legend.

R2-C3

3) As it stands, there is not enough direct evidence provided to support the model that the Shu complex regulates Rad53. The language should be softened to "may regulate" or "leads to a decrease in Rad53 activity/activation" instead. Alternatively, more experimental evidence should be provided. An alternative hypothesis is that if the Shu complex is disrupted and the cells are exposed to MMS, Rad53 is hyper-activated because the Shu complex isn't present to repair the damage. If Rad53 signaling is inhibited by the R605A mutation, then the Shu complex may not be needed because of the inability for the cell to detect the damage or perhaps the use of an alternative repair pathway to bypass the lesion (such as TLS).

Upon reflection, we completely agree with the reviewer. In the revised manuscript, we note that our spotting assays and western results show that deletion of the Shu complex leads to a decrease in Rad53 activity/activation, and now describe these results as being consistent either with Rad53 regulating the Shu complex, or that the Shu complex is important for damage sensing or alternative repair pathways. We softened our statement about the new role of the Shu complex as suggested by the reviewer (Pages 3, 9, 23 and 25).

R2-C4

4) The serial dilutions presented in Figure 4E and 4G are difficult to compare because of the differences observed between Shu complex mutants. In most cases where the Shu complex has been analyzed in vivo, individual Shu complex disruptions (SHU1, SHU2, CSM2, PSY3) leads to the same

phenotypes. Occasionally SHU1-SHU2 and PSY3-CSM2 will have different results (only observed in meiosis) but they are consistent with their binding partners.

a. In Figure 4E, *shu1Δ pph3Δ* cells are less sensitive than the other Shu complex members. It is possible that these differences have to do with incubation times or having results from different plates/controls. The serial dilutions need to be redone on a single plate for a more accurate comparison.

These are important questions. In the original submission all strains were incubated for the same period of time under the same conditions. For all measurement comparisons, strains had either been either grown on the same plate or on different plates with matched controls on each plates to provide confidence that conditions across plates were equivalent.

However, because of the potential importance of this concern, we repeated these assays with the relevant strains on a common plate (new Figure 4E), including Shu complex single mutants. We also took this opportunity to limit the possibility that suppressors had arisen in our original *shu1Δ pph3Δ* strain. To this end, we backcrossed *shu1Δ pph3Δ* with a BY4741 wild type strain, sporulated the heterozygous progeny, and dissected tetrads to obtain a new double mutant screening for G418 and clonNat resistance. Two independent isolates of each strain (single and double mutants) were used for sensitivity assays. Both recapitulated our original findings, showing *shu1Δ pph3Δ* to be slightly less sensitive to MMS than other *rad53-R605A* Shu complex members. We show a representative *shu1Δ pph3Δ* isolate in Figure 4E. The new spotting assay experiments thus reduce the chance that changes that the differences between double mutant observed in Figure 4E are due to differences in genetic background or growth condition.

We appreciate the suggestions, and the opportunity to strengthen our original finding that double mutants between genes encoding Shu complex members and *pph3Δ* are more sensitive to MMS than the underlying single-gene deletions (a situation that mirrors what was observed for double mutants between Shu complex genes and *slx4Δ*).

R2-C5

b. In Figure 4G, differences are seen this time with *rad53-R605A shu2Δ*. Since disruption of the Shu genes leads to increased mutation rates (due to TLS), it's possible that this double mutant has a suppressor and that is why there is better growth, which is comparable to a *rad53-R605A* single mutant. Again, because the suppression is difficult to observe except for the case of *rad53-R605A shu2Δ* (which looks different from the other Shu complex members), it important to redo the serial dilutions on the same plate or a few plates (with paired controls such as WT, *rad53-R605A*, *shu1Δ*, *rad53-R605A shu1Δ*, *shu2Δ*, *rad53Δ-R605A*, etc...).

As described above for the *shu1Δ pph3Δ* strain, we backcrossed our original *rad53-R605A shu2Δ* strain with a WT strain, sporulated, and selected two independent isolates each for single and double mutants. These were used to repeat the original sensitivity assays and both isolates recapitulated our original findings, showing that *rad53-R605A shu2Δ* is slightly less sensitive to MMS than other *rad53-R605A* combined with deletion of other Shu complex genes. We show representative results for a *rad53-R605A shu2Δ* isolate in Figure 4G. As suggested, we included single and double mutants side-by-side on each of two plates with WT, *RAD53* and *rad53-R605A* single mutants serving as matched controls across plates. The agreement of these newly generated strains suggests that the original results were not affected by unlinked suppressors in the original *shu1Δ pph3Δ* strain.

We again appreciate the suggestion and the opportunity to strengthen our previous results: double mutants involving *rad53-R605A* and a Shu complex gene are less sensitive to MMS than their single mutant counterparts, consistent with the idea that Shu complex deletions lead to decreased Rad53 activity/activation, or with the alternative hypotheses suggested by the reviewer.

Minor comments:

R2-C6

1) The sentence in the Discussion, p. 19 "Thus, our results provide evidence for a previously uncharacterized role of the Shu complex in the cellular response to DNA damage by MMS." is inaccurate statement. The Shu complex role has primarily been characterized upon MMS damage but the novel part provided is the link to Rad53.

We agree, and have replaced this sentence with the following one, which we hope the reviewer will agree is valid: "We detected and validated unanticipated interactions between *SLX4* and Shu complex genes, which mirrored the genetic interactions observed between *PPH3* and the Shu complex. We further found that presence of a functional Shu complex corresponded to reduced Rad53 activity during MMS treatment." (Page 24)

R2-C7

2) On p.17 the authors state "*SLX4* shows epistatic relationship with error-free lesion bypass genes during MMS treatment (Flott et al 2007)". Is the same phenotype observed in this study?

Flott et al (2007) showed that *SLX4* is epistatic both to *RAD6* and to *RAD18*, which each regulate error-free lesion bypass, i.e., deletion of *SLX4* did not further sensitize *rad6Δ* or *rad18Δ* mutants to MMS (Flott et al, Figure 2B). The authors did not test or report interactions between Shu complex genes and *slx4Δ*. Differences in the nature of interactions between *SLX4* and *RAD6/RAD18* from those between the *SLX4* and Shu complex genes is consistent with a role of the Shu complex that is independent of error-free bypass and seems also to be related to Rad53 activity.

R2-C8

3) On p. 13 Please mention the "10 culture conditions" in the main body or refer to Fig 3C for clarity.

We now refer to Fig 3C (Page16).

Reviewer #3:

R3-C1

Díaz-Medjía et al. introduce a new technique to generate yeast double mutants and subsequently assay them across multiple conditions. With respect to the numerous techniques already available, the proposed approach has the advantage to allow the generation of a single pool containing virtually all the desired double mutants. This means that such pool could be easily tested across many conditions with minimal labour and therefore with high-throughput. The authors tested the approach in a 34x38 gene combinations, and across 10 conditions related to DNA damage. They demonstrated how the technique could recapitulate previous experiments and genetic interaction screens, and went to further inspect a newly uncovered genetic interaction (*SLX4* and the Shu complex). The term "orgy mating" is surely catchy, and also the probable reference to a popular 90s videogame (BFG) has been noted.

The technique is sufficiently detailed so that the minute details can be easily inspected. Furthermore the authors included the original read counts and final genetic interaction score (GIS), allowing in-depth inspection of their proposed scoring scheme. Numerous other techniques are already available, including one that is conceptually very similar despite being more reliant on robotic work (iSeq, Jaffe et al., 2017). The authors fully acknowledge the existence of all these alternatives, and introduce compelling arguments why their own implementation can potentially streamline genetic-interaction screening across multiple conditions. The authors also acknowledge how their method shares the same weakness as iSeq, namely the development of aneuploidies when using the *can1* locus in generating the recipients, and they indicate how this could be fixed in future developments of the technique.

There are a number of issues with this manuscript, although none particularly serious.

We appreciate the thoughtful comments. We hope that the term BFG will remind most readers of the Big Friendly Giant from "BFG", the children's book by Roald Dahl. However, we cannot deny that, for some readers, the term will evoke a powerful weapon from the classic videogames Doom and Quake.

Although we also appreciate support for the term “orgy mating”, we have bowed to other editorial and reviewer comments and suggested replacement of this term with “party mating”. However, we would be happy to revisit this decision at the editor’s discretion.

R3-C2

I wasn't able to reproduce the GIS from the read counts provided, even if to the best of my knowledge I have implemented the author's same GIS scoring scheme. It seems that the authors have carried out some form of quantile normalization to their raw GIS, despite not reporting it in the method's section.

We regret that the analysis methods used in our original submission were not described well enough for Reviewer #3 to recapitulate our results. However, we’d like to clarify that we didn’t perform any quantile normalization.

In the meantime, we brought in another author (ACe), who was able to reproduce our original submission scores, and then substantially improved our approach to compute GIS and to estimate and control False Discovery Rates (FDR) for genetic interaction score (described in our response to R1-C1 above and below in R3-C3). We now provide all code as *Computer Code EVI*, as well as a GitHub link, so that the reviewer (and any reader) can check the analysis directly and run the code to reproduce the results.

After looking at supplementary figure S4 I think the problem might be related to the partition of the tested genes into "neutral" and "related to DNA damage", even though how this separation comes into play into the GIS computation is evident only for the wild-type fitness calculation.

The neutral gene set is used in three ways. First, as the reviewer notes, it is used to estimate wild-type fitness. Second, it is used to estimate single mutant fitness values (calculated as the mean of all strains harboring both the mutation of interest and a mutation in a neutral gene). By using only strains that carry two deletions, we achieve a constant genetic background such that each strain has the deletion-associated kanMX and natMX cassettes, Third, we now use the neutral set to calculate FDR for genetic interactions within each condition, as well as FDR for genetic interaction differences between conditions. We have completely re-written our methods for GIS computation and related analyses, and hope that the reviewer will find the updated description in the Materials and Methods section (starting Page 36) more clear.

R3-C3

This discrepancy also didn't allow to verify whether an absolute z-score cutoff of 1 is appropriate. The authors could provide a more detailed description of their scoring scheme or ideally a code implementation of their scoring procedure.

In the revised manuscript, we have substantially improved our interaction calling method. Now, all thresholds are explicitly chosen to control the false discovery rate (FDR) below a threshold of 1% (see Materials and Methods, Page 39). Note that the FDR procedure only controls false positives due to random error in our measurements, and disagreements with previously published studies may stem, for example, from strain differences between the studies, biological differences between liquid and solid growth conditions, or from false negatives or false positives from either random or systematic error in the previous study. However, despite the great potential for systematic differences, benchmarking the performance of BFG-GI interactions using a 1% FDR threshold against those in St. Onge et al. 2007 (with a permissive effect size cutoff of $|GIS| > 0.075$) yielded 77% precision for positive interactions, and 91% precision for negative interactions, at 44% and 64% sensitivity, respectively (Page 17).

All methods for scoring and FDR analysis are described in the Materials and Methods, with code now provided as *Computer Code EVI*.

R3-C4

Regarding the reproducibility of the approach: the authors have decided to merge the two technical replicates after noticing that the "relative strain abundance" (I'm assuming equivalent to Fijk as reported in Materials and methods) correlation was above 0.95. I would be interesting in knowing

as well the correlation in GIS. Read counts are bound to be highly correlated, so I'm not sure how that measure is really informative with respect to reproducibility.

In the revised manuscript, we have opted not to sum the counts of the two technical replicates. We now calculate GIS separately for each replicate, and average the resulting GIS scores into a single GIS score. The technical replicates are used to calculate an expected error for double mutant fitness measurements in each condition. For reference, we obtain the following correlations for GIS between the replicates before averaging the scores:

Condition	r
NoDrug	0.96
DMSO	0.97
MMS	0.96
4NQO	0.95
BLMC	0.94
ZEOC	0.95
HYDX	0.96
DXRB	0.96
CSPL	0.96
Over all conditions	0.96

The revised manuscript now notes that correlation of GIS between technical replicates of is $r = 0.96$ (Page 13).

R3-C5

Related to the previous point is also the choice of plotting the (raw?) colony size against read counts as a measure of comparability between "conventional" SGA and the proposed new method (figure 2A). Wouldn't a plot comparing GIS and something like an S-score be more appropriate?

The revised manuscript now makes clear (Page 11) that the purpose of this analysis was to assess the extent to which quantifying growth via fused-barcode-sequencing of pooled strains could recapitulate the measurements of growth in individual patches (as in conventional SGA). For this purpose, assessing the correlation between the two proxy measurements of cell count (number of pixels for cell patch growth assays and number of reads for competitive growth assays) seemed reasonable.

Proper calculation of S-scores would have required that we arrange plates such that there is a plate for every gene in our screening matrix, and that every plate have a particular mutant in common across the plate. We did not do this, as it was not required for the question we sought to address.

R3-C6

The authors have compared their approach to a previous study (St Onge et al., 2017) in two conditions. The scatter plots comparing the epsilon score with the GIS seem to indicate that the two scoring schemes have very different dynamic ranges; this is particularly evident when looking at the "no drug" condition. Can the authors comment on these differences?

We agree that the GI scores calculated in our previous manuscript exhibited a different dynamic range than the Epsilon computed by St Onge *et al.*, and that this was particularly apparent for the 'no drug' condition. We have since updated our analysis procedure so that growth rates are explicitly estimated for both single and double mutants (see response to R1-C1). As this is more similar to the growth rate estimation carried out by St Onge *et al.*, this should (and does) yield GIS values that are more comparable to Epsilon (Fig. 2 D-E).

Although the dynamic ranges are now similar for the no drug condition, they still differ for the MMS condition. Under the MMS condition, negative genetic interactions from our study seem to be more extreme than in St. Onge *et al.*, while positive interactions are less so. We ascribe this phenomenon to differences in MMS activity at the doses used in the two studies. MMS, like other reactive compounds, is known to be variable in activity between lots and over time, so that experimental doses are often set relative to minimal inhibitory concentrations that are measured shortly before each experiment. In addition, the drug diffusion and activity dose may be different between the liquid media used by St Onge *et al.*, and the solid Bioassay plates we used in BFG-GI. That our single-mutant fitness values are systematically higher than those measured in St Onge *et al.* suggests that we used MMS at a lower-activity dose. Single-mutant fitness values in MMS range from 0.2 - 1 in St. Onge *et al.*, as compared with a 0.64 - 1 range in our study. At drug doses with lower activity, we would expect negative genetic interactions to yield more negative epsilon values than at doses with higher activity (i.e. the faster the expected growth, the easier it is to detect slower-than-expected growth). Similarly, lower-activity doses should yield lower epsilon values for positive genetic interactions than high-activity doses (i.e. the faster the expected growth, the more difficult it is to detect faster-than-expected growth).

R3-C7

In previous studies of conditional genetic interactions (example: Bandyopadhyay et al, 2010) it was noted that the variance of the difference between genetic interaction scores depends on the magnitude of this difference. In other words, the most extreme genetic interaction scores tend to have higher variance across replicates than lower values. In these studies the difference in genetic interactions was therefore taking into account this dependency between variance across replicate and magnitude in difference in genetic interaction scores. This resulted in a somewhat non-intuitive set of conditional genetic interactions across conditions that was defined as a neutral genetic interaction in both condition. Would this also apply to the genetic interaction scores as defined in this study?

Interesting question. We tested for this phenomenon in our data, using modeled error for each GIS measurement. Like Bandyopadhyay *et al.*, we found that the magnitude of Δ GIS is correlated with the modeled error; however, the effect is very small ($r = 0.1$).

We agree with the reviewer that differential interactions that are not changes in interaction type (negative, neutral or positive) are non-intuitive and difficult to interpret. Although we provide information from all differential interaction tests for the reader, should they wish to consider within-type differential interactions, we now call differential interactions only when they change interaction type.

R3-C8

In figure 2F the authors use same gene pairs and linked genes to show how those gene pairs GIS are enriched in negative values. Why is the "same gene pairs" curve showing a shoulder above zero?

We thank the reviewer for bringing this to our attention. When investigating this shoulder, we found that it corresponds to strains which have a low abundance in the heterozygous diploid pool, which we used as time zero. The GIS measure requires that strains have a well-measured initial frequency in this pool in order to reliably estimate haploid double mutant fitness. Specifically, we found that same-gene strains (which in diploid strains had only one copy for the gene in question) for which $C_{xy} < 30$ in the diploid pool in either technical replicate appear to have GISs close to neutrality, instead of negative as expected. This is because if C_{xy} is low in the diploid stage, regardless if their C_{xy} is low also the haploid pools, they will tend to yield GIS values that are lower-confidence or appear as neutral. This is in part due to the addition of a pseudocount, which serves to 'conservatively nudge' less-well-measured double mutant towards neutrality. We now provide Fig EV4B to illustrate this behaviour.

We decided to remove strains having $C_{xy} < 30$ in the ‘heterozygous diploid pool’. This included 20 same-gene strains causing the shoulder in our previous submission and 52 different-gene strains (presumably complex-haplo-insufficient cases) out of a total of 3,220 different-gene strains measured.

R3-C9

The authors report that the double barcode total size is 325bp and how they have used an Illumina NextSeq PE platform. What was the read length? Is it sufficient to reliably distinguish pairs of barcodes? Would that be true if the number of tested genes was way higher? To what extent can such this method to scale up, eventually to a genome-wide level? The authors report how they optimised their mating setup for their 34x38 gene matrix, but it would perhaps be appropriate if they could add even just a theoretical account on the upper limit in terms of gene pairs that can be generated with their approach.

We thank the reviewer for this question. In our original submission we only noted 325bp as the size of the amplicon that contains the fused-barcode locus, but never actually specified the length of the barcodes themselves. In fact, we used 20 bp barcodes. The number of unique barcodes of this length is $4^{20} = 1.09 \times 10^{12}$, so that there is no shortage of complexity in this design to represent the ~5500 protein-coding genes in the yeast genome with several replicates per gene. For sequencing, we used a 75 cycle NextSeq kit and used 25 cycles for each barcode and 6 cycles for the multiplexing tag. The revised manuscript clarifies this in Materials and Methods (Page 34).

As for optimization of culture conditions, in fact, we optimized both mating and sporulation steps, not only for 34x38 gene matrix, but in general, to reduce potential bottlenecks. We calculated that to cover a yeast genome-scale matrix of 5,500 x 5,500 genes, with 1,000 representative cells for each cross, we would need $\sim 3 \times 10^{10}$ cells at each step along the BFG-GI procedure. We calculated that using the optimal conditions that we established for mating (22%) and sporulation (18%), we would need to culture pools in ~27 Bioassay 500cm² dishes for mating and ~10 L of liquid media for sporulation to cover all 5,500 x 5,500 crosses. Thus, in principle, BFG-GI could be extended to genome-scale studies. We have added this calculation in the Discussion (Page 26).

Minor comments:

R3-C10

There is a section of discussion (paragraph 6) that seems could be moved either in the results or the materials and methods section, or at least part of it.

Good suggestion. This has been moved to Materials and Methods (Pages 30 and 32) and replaced with discussion on the feasibility of extending BFG-GI to genome scale (Page 26).

R3-C11

In figure 3F a gene (RAD57) can be seen switching the direction of its genetic interaction quite often (in 13 condition pairs). Is this an unexpected result?

Given the revised the data analysis methods, *RAD57* participates in 211 differential between-condition interactions, 8 of which involve a sign reversal. We now show the distribution of differential interactions in Fig EV5 B-C, which shows that the *RAD57* pattern is not an outlier.

However, we now note that *RAD5* is an outlier in that it participates in 233 differential interactions, 55 of which involve a sign reversal. Moreover, we see interesting patterns in *RAD5* differential interactions with *MMS4*, *MUS81*, *RAD51*, *RAD54* and *RAD55* (Page 18), and we thank the reviewer for pointing us in this direction.

R3-C12

How many of the replicates with GIS $R < 0.5$ are recipients? An absolute number would be more accurate.

They were 15 out of 21. This is now mentioned on Page 13.

2nd Editorial Decision

23 April 2018

Thank you again for sending us your revised manuscript. We have now heard back from reviewer #3 who was asked to evaluate the revised study. As you will see below, reviewer #3 is now satisfied with the modifications made and thinks that the study is suitable for publication.

Before we formally accept the study for publication, we would like to ask you to address some remaining editorial issues listed below.

REVIEWER REPORT

Reviewer #3:

The authors have addressed the concerns raised by the reviewers. In particular they have made improvements to the statistical analysis of the genetic interactions and differential genetic interactions and have also improved the follow up study of the Shu complex interactions. I have no further concerns.

Corresponding Author Name: Frederick P. Roth

Manuscript Number: MSB-17-7985